# Composition of Triterpene Glycosides of the Far Eastern Sea Cucumber *Cucumaria conicospermium* Levin et Stepanov; Structure Elucidation of Five Minor Conicospermiumosides A_3_-1, A_3_-2, A_3_-3, A_7_-1, and A_7_-2; Cytotoxicity of the Glycosides Against Human Breast Cancer Cell Lines; Structure–Activity Relationships

**DOI:** 10.3390/md22120560

**Published:** 2024-12-16

**Authors:** Alexandra S. Silchenko, Ekaterina A. Chingizova, Ekaterina S. Menchinskaya, Elena A. Zelepuga, Anatoly I. Kalinovsky, Sergey A. Avilov, Kseniya M. Tabakmakher, Roman S. Popov, Pavel. S. Dmitrenok, Salim Sh. Dautov, Vladimir I. Kalinin

**Affiliations:** 1G.B. Elyakov Pacific Institute of Bioorganic Chemistry, Far Eastern Branch of the Russian Academy of Sciences, Pr. 100-letya Vladivostoka 159, 690022 Vladivostok, Russia; chingizova_ea@piboc.dvo.ru (E.A.C.); ekaterinamenchinskaya@gmail.com (E.S.M.); zel@piboc.dvo.ru (E.A.Z.); kaaniw@piboc.dvo.ru (A.I.K.); popov_rs@piboc.dvo.ru (R.S.P.); paveldmt@piboc.dvo.ru (P.S.D.); 2A.V. Zhirmunsky National Scientific Center of Marine Biology, Far Eastern Branch, Russian Academy of Sciences, Palchevskogo Str., 17, 690022 Vladivostok, Russia

**Keywords:** *Cucumaria conicospermium*, Dendrochirotida, triterpene glycosides, conicospermiumosides, sea cucumber, cytotoxic activity, human breast cancer, QSARs

## Abstract

Five new non-holostane di- and trisulfated triterpene pentaosides, conicospermiumosides A_3_-1 (**1**), A_3_-2 (**2**), A_3_-3 (**3**), A_7_-1 (**4**), and A_7_-2 (**5**) were isolated from the Far Eastern sea cucumber *Cucumaria conicospermium* Levin et Stepanov (Cucumariidae, Dendrochirotida). Twelve known glycosides found earlier in other *Cucumaria* species were also obtained and identified. The structures of new compounds were established on the basis of extensive analysis of the 1D and 2D NMR spectra, as well as by the HR-ESI-MS data. The aglycones of **1**–**5** differed by side chains structures. Additionally, conicospermiumoside A_7_-1 (**4**) had a 9(11)-double bond in the aglycone, while the remaining glycosides contained a 7(8)-intranuclear double bond. Eight types of carbohydrate chains known earlier from the glycosides of the sea cucumbers of the *Cucumaria* genus were found as part of the glycosides of *C. conicospermium*. The set of sugar chains of the glycosides from *C. conicospermium* was similar to that from *C. okhotensis*. The raw biogenetic series of aglycones, leading to the formation of hexa-nor-lanostane derivatives in the process of biosynthesis and a sort of functionally-structural division that was realized due to separation of biosynthetic pathways of holostane and lanostane derivatives, can be traced when the structures of the glycosides isolated from *C. conicospermium* are compared. The cytotoxic action against three human breast cancer cell lines (MCF-7, T-47D, MDA-MB-231), and non-tumor MCF-10A and hemolytic activity of compounds **1**–**5**, as well as seven known glycosides were tested. Conicospermiumosides A_3_-3 (**3**) and A_7_-1 (**4**), having a 22-oxo-23(24)-en fragment, were strongly hemolytic despite lacking a lactone in their aglycones. Moreover, both compounds demonstrated a promising suppressing action against triple negative breast cancer cells. The cells of the MDA-MB-231 line were most sensitive to the cytotoxic action of the glycosides, while the MCF-7 cell line was most sustainable. Six glycosides were selected for further study of some aspects of anticancer action against MDA-MB-231. The selective action of the compounds **4** and **8** on the MDA-MB-231 cells without significant toxicity against the MCF-10A cells was noticeable. More importantly, the selectivity of the compounds was changed over time and maximal selectivity to cancer cells was demonstrated by glycoside **1** at 48 h of exposition. The glycosides **1**, **3** and the desulfated derivative **7a** strongly inhibited colony formation and growth of the TNBC cells until the process stops completely. Okhotoside B_1_ (**8**), DS-okhotoside A_1_-1 (**7a**), and conicospermiumoside A_3_-3 (**3**) showed a potent cell migration-inhibiting capacity. Quantitative structure–activity relationships (QSARs) calculated on the basis of a correlational analysis of the physicochemical properties and structural features of the glycosides and their cytotoxic activity against different cell lines showed some structural features influenced differently, sometimes even in opposite ways, on the activity of glycosides toward diverse cells (erythrocytes, MCF-10A, and TNBC MDA-MB-231 cells). This observation indicated that glycosides obviously target different membrane components, such as lipids of erythrocytes and some receptors on the surface of mammary normal or tumor cells.

## 1. Introduction

Marine invertebrates of the class Holothuroidea (phylum Echinodermata) are called sea cucumbers or holothurians. The class consists of 8 orders, including the most numerous in terms of the number of species, the order Dendrochirotida. The representatives of this order are nowadays most often the objects of chemical research since they contain the variable triterpene glycosides—secondary metabolites possessing pharmacological potential [1,2]. As the glycosidic composition in the sea cucumbers is usually rather complex, attempts to obtain pure compounds have stimulated the modernization of separation approaches of the mixtures of native metabolites [3].

Early studies of different representatives of the genus *Cucumaria* (family Cucumariidae, order Dendrochirotida) showed their glycosides characterized by species-specific sets of aglycones and a genus-specific set of carbohydrate chains including mono-, di-, and trisulfated pentaosides, branched by the second monosaccharide residue [4]. Then, the set of sugar moieties inherent for *Cucumaria*’s glycosides was broadened with mono-, di-, and trisulfated tetrasaccharide chains found in the glycosides of *C. frondosa* [5], *C. okhotensis* [6], and *C. djakonovi* [7,8]. However, the branched pentasaccharide chains were always present and, as a rule, were parts of predominant glycosides from the glycosidic fractions [4,9].

Extensive studies of the distribution and ecology of the sea cucumbers inhabiting seas of the Far East of Russia demonstrated significant species diversity. Several new species of the genus *Cucumaria* have been described: *C. djakonovi*, *C. savelijevae*, *C. conicospermium* [10], *C. levini*, *C. anivaensis* [11], and *C. okhotensis* [12]. Whereas previously all these species were identified as one species—*C. japonica*, which turned out to be polyphyletic. Actually, the areal of *C. japonica* is significantly narrower than it was considered earlier. Given the fact that some sea cucumbers species of the genus *Cucumaria* are commercially important, their exact classification and identification are very important.

The difficulty of definition of *Cucumaria* species based on morphological features only is due to high variability of the ossicles, including age-related variability. In this case the use of triterpene glycosides as chemotaxonomic markers becomes relevant. The successful examples of application of knowledge on chemical structures of the glycosides for the holothurian’s systematics are as follows: *C. frondosa* [5,13,14,15,16], *C. conicospermium* [9], *C. okhotensis* [6,17] and *C. djakonovi* [7,8]; those metabolites differ significantly from those of *C. japonica*.

Considering the fact that sea cucumber glycosides possess pharmacological activity, knowledge of differences in the chemical composition of commercial species of *Cucumaria* is relevant for standardization of the raw material.

For the glycosides isolated from the representatives of the genus *Cucumaria*, the anticancer activity against different types of tumors is known [18,19,20,21,22,23,24,25,26,27,28,29]. Recently obtained data indicate that glycosides from *C. djakonovi* could be used in the target therapy of triple-negative breast cancer [30]. Thus, as chemical investigations of the sea cucumbers of the *Cucumaria* genus continue to search for new minor glycosides and compounds active against human breast cancer, the glycosidic composition of *Cucumaria conicospermium* Levin et Stepanov was studied, including the isolation, structure elucidation, and studying the cytotoxic action of the glycosides against three breast cancer cell lines (MCF-7, T-47D, and MDA-MB-231) and normal MCF-10A cells, as well as erythrocytes. Anti-proliferating activity, the ability to inhibit colony formation and growth, and to stop the migration of cancer MDA-MB-231 cells was investigated for six glycosides. Analysis of the quantitative structure–activity relationships for the glycosides of *C. conicospermium* (including two desulfated derivatives) allowed us to assess the significance of separate chemical features for the bioactivity against different cell lines.

The composition of glycosides of *C. conicospermium*, collected near Zolotoy Point at the Western shore of the Sea of Japan, was investigated earlier [9], but the constituents of polar fractions (di- and trisulfated glycosides) were isolated as desulfated derivatives due to the difficulties of their separation as native sulfated compounds at that time. The structures of five glycosides have been elucidated, including native monosulfated cucumarioside A_2_-5, disulfated cucumariosides A_3_-2 and A_3_-3 (as desulfated derivatives) with hexa-nor-holostane aglycones isomeric by an intranuclear double bond position and trisulfated koreoside A and isokoreoside A, having the same pair of aglycones.

The samples of *C. conicospermium* for the current research were collected near Onekotan Island (the Kurile Islands) during 36th research cruise of the research vessel “Akademik Oparin”. The isolation, structure elucidation, and biological activity of five new minor glycosides, conicospermiumosides A_3_-1 (**1**), A_3_-3 (**2**), A_3_-3 (**3**), A_7_-1 (**4**), and A_7_-2 (**5**), as well as twelve known compounds are reported. The chemical structures of **1**–**5** were established by the analyses of the ^1^H, ^13^C NMR, 1D TOCSY, and 2D NMR (^1^H,^1^H COSY, HMBC, HSQC, ROESY) spectra as well as the HR-ESI mass spectra. All the original spectra are displayed in Appendix A.

## 2. Results

### 2.1. Structure Elucidation of Glycosides

As a result of the separation of glycosidic sum of *C. conicospermium*, five new glycosides were isolated from the two most polar fractions: disulfated pentaosides—conicospermiumosides A_3_-1 (**1**), A_3_-2 (**2**), and A_3_-3 (**3**); and trisulfated pentaosides—conicospermiumosides A_7_-1 (**4**) and A_7_-2 (**5**), as well as known frondoside A_7_-4 (**14**), djakonovioside F_1_ (**15**), koreoside A (**16**), and isokoreoside A (**17**). Three less polar fractions contained only known monosulfated glycosides with tetra- and pentasaccharide chains—djakonovioside A (**6**), okhotoside A_1_-1 (**7**), okhotoside B_1_ (**8**), frondoside A (**10**), okhotoside A_2_-1 (**11**), cucumarioside A_2_-5 (**12**), and djakonovioside B_2_ (**13**), as well as disulfated tetraoside okhotoside B_2_ (**9**) (Figure 1).

The ^13^C NMR spectra of carbohydrate parts of conicospermiumosides A_3_-1 (**1**) (Table 1, Appendix A), A_3_-2 (**2**) (Appendix A), and A_3_-3 (**3**) (Appendix A) coincided to each other indicating the presence of identical carbohydrate chains consisting of five residues and containing sulfate groups at the first and third sugar units. The same carbohydrate moiety was found first in cucumarioside A_3_ isolated from the sea cucumber *C. japonica* [31]. The ^1^H, ^13^C NMR, and HSQC spectra of the carbohydrate moiety of conicospermiumoside A_3_-1 (**1**) demonstrated five doublets of the anomeric protons at δ_H_ 4.72–5.21 (*J* = 6.9–8.3 Hz) and five signals of anomeric carbons at δ_C_ 102.2–105.2 indicating the presence of the pentasaccharide chain with *β*-glycosidic bonds. The algorithm of analysis of carbohydrate chain structure was as follows: the finding of anomeric proton of each sugar residue by the ROESY and 1D TOCSY spectra, analysis of the ^1^H, ^1^H COSY correlations, subsequent analysis of the ROESY and HSQC spectra that resulted in the establishing of the type of each monosaccharide, followed by analysis of the NOE- and HMBC correlations that allowed us to determine the positions of glycosidic bonds (β(1→2), β(1→3), and β(1→4)). Hence, it was found that first monosaccharide residue of **1** (Xyl1) was attached to C-3 of the aglycone and sulfated by C-4; the second sugar (Qui2) was linked to the C-2 Xyl1 and contained two branchpoints; the third sugar (Glc3) was attached to C-4 Qui2 and contained sulfate group at C-6; the fourth residue (MeGlc4) bonded to C-3 Glc3 and was terminal; another terminal unit was Xyl5, linked to C-2 Qui2.

The positions of sulfate groups were deduced from the observed α- and β-shifting effects: the signal of C-4 Xyl1 shifted downfield to δ_C_ 76.1, while the signals of C-3, 5 Xyl1 shifted upfield to δ_C_ 75.0 and 64.0, respectively; the signal of C-6 Glc3 deshielded to δ_C_ 67.2, whereas the signal of C-5 Glc3 shielded to δ_C_ 74.9, indicating sulfate groups were attached to C-4 Xyl1 and C-6 Glc3.

The molecular formula of conicospermiumoside A_3_-1 (**1**) was determined as C_59_H_96_O_31_S_2_Na_2_ from the [M_2Na_ − Na]^−^ ion peak at *m*/*z* 1387.5269 (calc. 1387.5280) and [M_2Na_ − 2Na]^2−^ ion peak at *m/z* 682.2704 (calc. 682.2694) in the (−)HR-ESI-MS (Appendix A), corroborating the presence of two sulfate groups. Glycoside **1** contained non-holostane aglycone without a lactone (signal of methyl group C-18 was observed at δ_C_ 25.0 and signal of quaternary hydroxylated carbon C-20 was observed at δ_C_ 76.9) with the intranuclear 7(8)-double bond (the signals of CH-7 at δ_C_ 122.2 and δ_H_ 5.65 (m) and C-8 at δ_C_ 148.9) and the 24(25)-double bond in the side chain (the signals of CH-24 at δ_C_ 123.1, δ_H_ 5.49 (H-24, brt (*J* = 7.3 Hz) and C-25 at δ_C_ 132.2) (Table 2). Another deshielded signal was assigned as the C-22 bearing hydroxyl group (δ_C_ 77.0 (C-22), δ_H_ 3.73 dd (H-22, *J* = 3.5; 8.6 Hz)). The configuration of the C-22 asymmetric center was deduced as *R* based on the coincidence of the values of chemical shifts of C-20 and C-22 in **1** that is characteristic for 5*α*-cholestan-3*β*,20*R*,22*R*-triol stereoisomer [32]. The same configuration of C-22 was deduced earlier for frondoside C having lanostane aglycone without lactone and 22-O-acetoxy-group [13]. The NOE-correlation H-22/H-16 (Table 2) confirmed the *R*-configuration of C-22. The same orientation of the 22-hydroxyl group is inherent for all known glycosides [7] probably due to the its significance for the biosynthesis of hexa-*nor*-lanostane aglycones like in koreoside A (**16**) and isokoreoside A (**17**). The aglycone of conicospermiumoside A_3_-1 (**1**) was the same as in frondoside A_7_-4 found first in the sea cucumber *C. frondosa* [16] and also isolated and identified in *C. conicospermium*.

The (−)ESI-MS/MS of **1** (Appendix A) demonstrated the fragmentation of the [M_2Na_ − Na]^−^ ion, with *m*/*z* 1387.5 giving fragment ion peaks at *m*/*z* 1267.5 [M_2Na_ − Na − NaHSO_4_]^−^, 1255.5 [M_2Na_ − Na − Xyl]^−^, 1153.5 [M_2Na_ – Na − NaSO_3_ − Xyl + H]^−^, 815.4 [M_2Na_ − Na − Agl − Xyl]^−^. The fragmentation of the [M_2Na_ + Na]^+^ ion (*m*/*z* 1433.5) was deduced from (+)ESI-MS/MS of **1** where the ion peaks with *m*/*z* 1331.6 [M_2Na_ + Na − NaSO_3_]^+^, 1301.6 [M_2Na_ + Na − Xyl + H]^+^, 1241.5 [M_2Na_ + Na − MeGlc + H]^+^, 1097.6 [M_2Na_ + Na − Xyl − 2NaSO_3_ + H]^+^, 1023.5 [M_2Na_ + Na − Xyl − NaSO_3_ − MeGlc + H]^+^, 873.2 [M_2Na_ + Na − Agl − NaSO_3_ + H]^+^, 741.1 [M_2Na_ + Na − Agl − XylSO_3_ + H]^+^, and 565.1 [M_2Na_ + Na − Agl − XylSO_3_ − MeGlc + H]^+^ were observed. The MS data confirmed both the aglycone structure and the sequence of monosaccharides in the carbohydrate chain of **1**.

The sugar absolute configuration for **1** and all other glycosides was assigned as D, as in all other sea cucumbers, along with biogenetic reasons, since the D-configuration of monosaccharides was determined earlier for okhotoside A_1_-1 (**7**) after the acidic hydrolysis of native glycoside and the subsequent alcoholysis of the monosaccharide mixture by the action of (R)-(–)-2-octanol, followed by acetylation, and GLC analysis in the presence of octyl-derivatives of corresponding standards of D and L monosaccharides. Glycoside **7** was isolated first from *Cucumaria okhotensis* [17] and also found by us in *C. conicospermium*.

These data indicate that conicospermiumoside A_3_-1 (**1**) is 3β-*O*-{3-*O*-methyl-β-D-glucopyranosyl-(1→3)-6-*O*-sodium sulfate-β-D-glucopyranosyl-(1→4)-[(1→2)-β-D-xylopyranosyl]-β-D-quinovopyranosyl-(1→2)-4-*O*-sodium sulfate-β-D-xylopyranosyl}-20*R*,22*R*-dihydroxylanosta-7,24-diene.

The molecular formula of conicospermiumoside A_3_-2 (**2**) was determined as C_61_H_98_O_32_S_2_Na_2_ from the [M_2Na_ − Na]^−^ ion peak at *m*/*z* 1429.5381 (calc. 1429.5386) and the [M_2Na_ − 2Na]^2−^ ion peak at *m*/*z* 703.2761 (calc. 703.2747) in the (−)HR-ESI-MS (Appendix A). The non-holostane nucleus of the aglycone of **2** (Table 3, Appendix A) was identical to that of **1**, the difference was in the side chain structure. The signal of C-22 (δ_C_ 79.6) was deduced from the characteristic correlation H-21/C-22 in the HMBC spectrum of **2**. The signal of the corresponding proton H-22 was observed at δ_H_ 5.19 (dd, *J* = 4.1; 9.2 Hz) indicating its deshielding in the spectrum of **2** as compared to that of **1**. The correlation between H-22 and the signal of quaternary carbon at δ_C_ 172.3 (OCOCH_3_) in the HMBC spectrum of **2** indicated the bearing of C-22 to the *O*-acetic group. The molecular weights difference in the MS spectra of **1** and **2** of 42 *amu* confirmed the presence of this functionality in **2**. Additional characteristic signals in the ^13^C and ^1^H NMR spectra of **2** at δ_C_ 121.0 (C-24), δ_H_ 5.24 (brt, *J* = 6.7 Hz, H-24), and 133.5 (C-25) corresponded to the 24(25)-double bond, that was confirmed by the correlations H-26(27)/C: 24, 25, 27(26) in the HMBC spectrum. The configuration of the C-20 and C-22 chiral centers in **2** was assigned as *R*,*R* based on biosynthetic background since the aglycone of conicospermiumoside A_3_-2 (**2**) proved to be identical to that of the isofrondoside C found first in *C. frondosa* [16] and confirmed by H-22/H-16, H-17 NOE-correlations.

The fragment ion peaks in the (−)ESI-MS/MS of **2** (Appendix A) were observed at *m*/*z* 1369.5 [M_2Na_ − Na − CH_3_COOH]^−^, 1249.5 [M_2Na_ − Na − NaHSO_4_ − CH_3_COOH]^−^ and in (+)ESI-MS/MS − at *m*/*z* 1415.5 [M_2Na_ + Na − CH_3_COOH]^+^, 1355.5 [M_2Na_ + Na − NaHSO_4_]^+^, 1295.5 [M_2Na_ + Na − NaHSO_4_ − CH_3_COOH]^+^, 1223.5 [M_2Na_ + Na − NaHSO_4_ − Xyl]^+^, 991.7 [M_2Na_ + Na − Agl]^+^, 873.2 [M_2Na_ + Na − Agl − NaSO_4_]^+^, and 741.1 [M_2Na_ + Na − Agl − XylSO_3_]^+^, being in good accordance with the NMR data and confirming the structure of conicospermiumoside A_3_-2 (**2**).

Thus, conicospermiumoside A_3_-2 (**2**) is 3β-*O*-{3-*O*-methyl-β-D-glucopyranosyl-(1→3)-6-*O*-sodium sulfate-β-D-glucopyranosyl-(1→4)-[(1→2)-β-D-xylopyranosyl]-β-D-quinovopyranosyl-(1→2)-4-*O*-sodium sulfate-β-D-xylopyranosyl}-20*R*-hydroxy,22*R*-acetoxylanosta-7,24-diene.

The molecular formula of conicospermiumoside A_3_-3 (**3**) was determined as C_59_H_94_O_31_S_2_Na_2_ from the [M_2Na_ − Na]^−^ ion peak at *m*/*z* 1385.5111 (calc. 1385.5124), the [M_2Na_ − 2Na]^2−^ ion peak at *m*/*z* 681.2622 (calc. 681.2616) in the (−)HR-ESI-MS (Appendix A). The ^13^C NMR signals of the aglycone part of **3** (Table 4, Appendix A) coincided with those of djakonovioside F_1_ (**15**) isolated for the first time from *C. djakonovi* [8] and also found now in *C. conicospermium*. The lanostane aglycone of **3**, lacking a lactone, contained the 7(8)- and 24(25)-double bonds as well as the 22-oxo-group in the side chain. The presence of the 22-oxo-24(25)-en fragment made the protons of the methylene group CH_2_-23 easily exchangeable to deuterium during the registration of the NMR spectra. For this reason, the signal of C-23 was not accumulated in the ^13^C NMR spectrum of **3**. The same situation was observed for djakonovioside F_1_ (**15**) [8]. To obtain the signal of C-23, the ^13^C NMR spectrum of **3** were repeatedly acquired in C_5_D_5_N/H_2_O that resulted in the appearance of the signal of C-23 at δ_C_ 36.9. The only difference between compounds **3** and **15** was in the quantity of sulfate groups. Actually, the comparison of their ^13^C NMR spectra demonstrated the coincidence of all the signals with the exception of the signals of C-6 and C-5 of MeGlc4. The first signal was observed at δ_C_ 61.7 in the spectrum of **3** but was deshielded to δ_C_ 67.1 in the spectrum of **15** due to the α-shifting effect of sulfate group attached to this position in **15** and absent in **3**. The other signal (C-5 MeGlc4) was shifted oppositely (δ_C_ 77.5 in **3** and δ_C_ 75.4 in **15**) due to the β-shifting effect of the sulfate group.

The (−)HR-ESI-MS of **3** (Appendix A) demonstrated ion peaks corresponding to monosulfated derivative formed as result of the loss of one sulfate group—[M_2Na_ − Na − SO_3_Na + H]^−^ ion at *m*/*z* 1283.5719 (calc. for C_59_H_95_O_28_SNa 1283.5736)—and to desulfated derivative formed as result of the loss of two sulfate groups—[M_2Na_ − Na − 2SO_3_Na + 2H]^−^ ion at *m*/*z* 1203.6159 (calc. for C_59_H_96_O_25_ 1203.6168). The fragment ion-peaks in the (−)ESI-MS/MS spectra were observed as a result of further fragmentation of the ion-peak at *m*/*z* 1283.5 and corresponded to the loss of terminal (xylose) residue, *m*/*z* 1151.5 [M_2Na_ − Na − SO_3_Na − Xyl + H]^−^, and subsequent loss of the aglycone moiety, *m*/*z* 695.2 [M_2Na_ − Na − SO_3_Na − Xyl − Agl]^−^. Fragmentation of the ion at *m*/*z* 1203.5 led to the consecutive detachment of monosaccharides; corresponding ion-peaks were observed at *m*/*z* 1027.5 [M_2Na_ − Na − 2SO_3_Na − MeGlc + H]^−^, 865.5 [M_2Na_ − Na − 2SO_3_Na − MeGlc − Glc + H]^−^, and 733.5 [M_2Na_ − Na − 2SO_3_Na − MeGlc − Glc − Xyl + H]^−^.

Thus, conicospermiumoside A_3_-3 (**3**) is 3β-*O*-{3-*O*-methyl-β-D-glucopyranosyl-(1→3)-6-*O*-sodium sulfate-β-D-glucopyranosyl-(1→4)-[(1→2)-β-D-xylopyranosyl]-β-D-quinovopyranosyl-(1→2)-4-*O*-sodium sulfate-β-D-xylopyranosyl}-20*R*-hydroxy,22-oxolanosta-7,24-diene.

The ^13^C NMR spectra of carbohydrate parts of conicospermiumosides A_7_-1 (**4**) (Table 5, Appendix A) and A_7_-2 (**5**) (Appendix A) coincided to each other due to the identity of their pentasaccharide chains containing three sulfate groups. The same carbohydrate moieties are characteristic for the glycosides of all representatives of the genus *Cucumaria* and named as the A_7_ group. This oligosaccharide chain was found first in the glycosides of the sea cucumber *C. japonica* [33]. The ^1^H, ^13^C NMR, and HSQC spectra of the carbohydrate moiety of conicospermiumoside A_7_-1 (**4**) demonstrated five doublets of the anomeric protons at δ_H_ 4.73–5.21 (*J* = 7.4–7.9 Hz) and five signals of anomeric carbons at δ_C_ 102.2–105.2 indicating the presence of five monosaccharide residues linked through *β*-glycosidic bonds. The sugar chains of **4** and **5** were characterized by the same monosaccharide composition as the compounds **1**–**3** differing by the presence of an additional (third) sulfate group attached to C-6 of terminal 3-*O*-methylglucose. Actually, the signals of C-5 and C-6 MeGlc4 were observed at δ_C_ 75.5 and δ_C_ 67.0, respectively, in the spectrum of **4** (Table 5). Positions of glycosidic bonds were typical for all known pentaosides branched by C-2 Qui2 [4,9,13,31,33].

The molecular formula of conicospermiumoside A_7_-1 (**4**) was determined as C_59_H_93_O_34_S_3_Na_3_ from the [M_3Na_ − 2Na]^2−^ ion peak at *m*/*z* 732.2305 (calc. 732.2310), and the [M_3Na_ − 3Na]^3−^ ion peak at *m*/*z* 480.4910 (calc. 480.4909) in the (−)HR-ESI-MS and [M_3Na_ + Na]^+^ ion peak at *m*/*z* 1533.4285 (calc. 1533.4296) in the (+)HR-ESI-MS (Appendix A). The aglycone of **4** was characterized by lacking a lactone (the signals of methyl group CH_3_-18 at δ_C_ 16.4 and δ_H_ 1.01 (s), the signal of C-20 at δ_C_ 81.6 in the ^13^C and ^1^H NMR spectra (Table 6, Appendix A)). The shielding of C-18 as compared to compounds **1**–**3** as well as the presence of signals at δ_C_ 148.2 (C-9) and 114.8 (C-11), δ_H_ 5.18 (m, H-11) indicated the 9(11)-position of the intranuclear double bond, instead of 7(8)- in the glycosides **1**–**3**. The deshielding of the signal of C-20 to δ_C_ 81.6 and the presence of the signal at δ_C_ 216.4 as well as the signals at δ_C_ 117.3 (C-24), δ_H_ 5.44 (m, H-24), and δ_C_ 135.0 (C-25) indicated the 22-oxo-24(25)-en fragment was inherent for **4** making its side chain identical to that of **3**. Noticeably, the signal of C-23 (δ_C_ 36.9) was found only after the registration of the ^13^C NMR spectrum of **4** in C_5_D_5_N/H_2_O. The aglycone of conicospermiumoside A_7_-1 (**4**) is found first and is isomeric to the aglycones of conicospermiumoside A_3_-3 (**3**) and djakonovioside F_1_ (**15**) by the position of the double bond in the polycyclic nucleus.

The fragment ion peaks in the (−)ESI-MS/MS of **4** (Appendix A) were observed at *m*/*z* 681.3 [M_3Na_ − 2Na − SO_3_Na + H]^2−^, 666.2 [M_3Na_ − 2Na − Xyl]^2−^, and 615.2 [M_3Na_ − 2Na − Xyl − SO_3_Na + H]^2−^, whereas in the (+)ESI-MS/MS of **4** they were observed at *m*/*z* 1431.5 [M_3Na_ + Na − SO_3_Na + H]^+^, 1413.5 [M_3Na_ + Na − NaHSO_4_]^+^, 1329.5 [M_3Na_ + Na − 2SO_3_Na + 2H]^+^, 1255.4 [M_3Na_ + Na − MeGlcSO_3_]^+^, 975.1 [M_3Na_ + Na − SO_3_Na − Agl]^+^, and 843.1 [M_3Na_ + Na − SO_3_Na − Agl − Xyl]^+^.

Thus, conicospermiumoside A_7_-1 (**4**) is 3β-*O*-{6-*O*-sodium sulfate-3-*O*-methyl-β-D-glucopyranosyl-(1→3)-6-*O*-sodium sulfate-β-D-glucopyranosyl-(1→4)-[(1→2)-β-D-xylopyranosyl]-β-D-quinovopyranosyl-(1→2)-4-*O*-sodium sulfate-β-D-xylopyranosyl}-(20*R*)-hydroxy-22-oxo-lanosta-9(11),24-diene.

The molecular formula of conicospermiumoside A_7_-2 (**5**) was determined as C_59_H_95_O_35_S_3_Na_3_ from the [M_3Na_ − 2Na]^2−^ ion peak at *m*/*z* 741.2344 (calc. 741.2362), and the [M_3Na_ − 3Na]^3−^ ion peak at *m*/*z* 486.4942 (calc. 486.4944) in the (−)HR-ESI-MS (Appendix A). The polycyclic nucleus of **5** was identical to those of conicosperiumosides of the group A_3_ (**1**–**3**) having a 7(8)-double bond (Table 7, Appendix A). The structure of side chain was elucidated starting from the correlation H-21/C-22 in the HMBC spectrum of **4**, indicating the signal of C-22 was observed at δ_C_ 78.1 that is characteristic for hydroxyl-bearing methine group (δ_H_ 4.29 (m, H-22)). This proton was correlated with two olefinic protons (signals at δ_C_ 6.13 (m, H-23 and H-24) assigned to the 23(24)-double bond (δ_C_ 126.2 (C-23) and δ_C_ 141.8 (C-24). In addition, these olefinic protons were correlated in the HMBC spectrum with the signal at δ_C_ 69.8 assigned as C-25. Actually, the signals of geminal methyl groups CH_3_-26 (δ_H_ 1.49, s, H-26) and CH_3_-27 (δ_H_ 1.48, s, H-27) were also correlated with this signal corroborating the presence of the 25-OH group in the side chain. The aglycone of conicospermiumoside A_7_-2 (**5**) is new due to the unusual structure of the side chain.

The fragment ion peaks in the (−)ESI-MS/MS of **5** (Appendix A) were observed as a result of the fragmentation of the ion [M_3Na_ − Na]^−^, giving the ions at *m*/*z* 847.4 [M_3Na_ − Na −MeGlcSO_3_ − GlcSO_3_ − Xyl + H]^−^, arising as a result of the sequential loss of monosaccharide units and 797.1 [M_3Na_ − Na − Agl − XylSO_3_ − H]^−^, 519.0 [M_3Na_ − Na − Agl − XylSO_3_ − MeGlcSO_3_ − H]^−^, corroborating the aglycone structure. Fragmentation of ion [M_3Na_ − 2Na]^2−^ led to the ion-peaks at *m/z* 690.3 [M_3Na_ − 2Na − NaSO_3_]^2−^ and 683.2 [M_3Na_ − 2Na − C_6_H_11_O_2_ − H]^2−^ arose as result of side chain detachment due to the cleavage of the 20(22)-covalent bond.

All these data confirmed the structure of conicospermiumoside A_7_-2 (**5**) as 3β-*O*-{6-*O*-sodium sulfate-3-*O*-methyl-β-D-glucopyranosyl-(1→3)-6-*O*-sodium sulfate*-*β-D-glucopyranosyl-(1→4)-[(1→2)-β-D-xylopyranosyl]-β-D-quinovopyranosyl-(1→2)-4-*O*-sodium sulfate-β-D-xylopyranosyl}-(20*R*,22*R,25*)-trihydroxylanosta-7,23-diene.

The structures of known compounds, including djakonoviosides A (**6**), B_2_ (**13**) [7], and F_1_ (**15**) [8], okhotosides A_1_-1 (**7**), A_2_-1 (**11**) [17], B_1_ (**8**), and B_2_ (**9**) [6], frondoside A (**10**) [34], cucumarioside A_2_-5 (**12**) [9], frondoside A_7_-4 (**14**) [16], koreoside A (**16**) [35], and isokoreoside A (**17**) [9], were identified by comparison of their ^13^C NMR spectra with the data in the literature. All the original spectra and the assignments of the signals are provided in Appendix A.

### 2.2. Biological Activity of the Glycosides

The cytotoxic activity of new glycosides **1**–**5** and some known compounds isolated from *C. conicospermium*, as well as desulfated derivatives **7a** (NMR spectrum Appendix A) and **8a** (NMR spectrum Appendix A) of okhotosides A_1_-1 (**7**) and B_1_ (**8**), respectively, formed spontaneously during the desalting of **7** and **8** by hydrophobic chromatography and then isolated by Si-gel CC, was studied against human erythrocytes and three types of human breast cancer cells (MCF-7, T-47D, and triple negative (TNBC) MDA-MB-231), as well as the non-tumor mammary epithelial cell line MCF-10A. Okhotoside A_1_-1 (**7**) and cisplatin were used as the positive controls. Cytotoxic activity against all the selected cell lines was assessed using the MTT method (Table 8).

Almost all tested compounds demonstrated strong hemolytic activity, with the exception of the moderately active conicospermiumoside A_3_-2 (**2**) (ED_50_ 7.56 ± 0.08) and the weakly active conicospermiumoside A_7_-2 (**5**) (ED_50_ 20.42 ± 0.36).

The cells of the MDA-MB-231 line were most sensitive to the cytotoxic action of the glycosides, while the MCF-7 cell line was most sustainable. Some of the compounds, like **4**, **8**, **10**, **14**, demonstrated selectivity being only slightly toxic against normal MCF-10A cells but highly cytotoxic in relation to TNBC cells. The maximal activity in the series against the MDA-MB-231 cell line was demonstrated by monosulfated glycosides **7** (IC_50_ 2.25 ± 0.17 μM) and **10** (IC_50_ 2.79 ± 0.09 μM) and desulfated compounds **7a** (IC_50_ 2.83 ± 0.36 μM) and **8a** (IC_50_ 1.09 ± 0.08 μM) (Table 8). Trisulfated conicospermiumoside A_7_-1 (**4**) was strongly hemolytic (ED_50_ 0.36 ± 0.04 μM) and almost non-active against normal MCF-10A cells (IC_50_ 40.25 ± 2.10 μM) but, at the same time, highly cytotoxic in relation to MDA-MB-231 cells (IC_50_ 4.78 ± 0.23 μM). The same tendency was observed for okhotoside B_1_ (**8**) but its action against the MDA-MB-231 cell line was less pronounced (IC_50_ of 11.35 ± 0.20 μM), and for desulfated derivative **8a** but it was more cytotoxic against all cell lines (IC_50_ (MDA-MB-231) 1.09 ± 0.08 μM and IC_50_ (MCF-10A) 9.87 ± 0.73 μM) demonstrating however almost a 10-fold difference in the action toward normal and cancer cells.

To study anti-proliferative properties, the four most prospective glycosides (**1**, **3**, **4**, **8**) and desulfated derivatives **7a** and **8a** were selected and prolonged incubation of cells for 48 and 72 h with glycosides was performed (Table 9, Appendix A). Conicospermiumosides A_3_-1 (**1**) and A_7_-1 (**4**) demonstrated weak cytotoxic action against normal MCF-10A cells during 48 h of exposition; glycoside **1** kept this property during 72 h; while glycoside **4** tripled its activity. Simultaneously, compound **1** was highly active against MDA-MB-231 cells at 48 h (IC_50_ 2.54 ± 0.09 μM) and moderately active at 72 h (IC_50_ 12.40 ± 0.75 μM); while compound **4** retained the activity against TNBC cells during 48 and 72 h at the same level (IC_50_ 15.29 ± 1.01 μM and IC_50_ 11.18 ± 0.73 μM). Conicospermiumoside A_3_-3 (**3**) and DS-okhotoside A_1_-1 (**7a**) demonstrated anti-proliferative properties saving its activity during 72 h at the same level as at 24 h (Table 8 and Table 9). Okhotoside B_1_ (**8**) increased the activity against MCF-10A cells by ~2.5 times at prolonged incubation as compared with 24 h of exposition (IC_50_ 40.95 ± 0.81 μM at 24 h, 15.54 ± 0.81 μM at 48 h, 16.49 ± 0.98 μM) but, at the same time, became more cytotoxic at 72 h in relation to MDA-MB-231 cells (IC_50_ 11.35 ± 0.20 μM at 24 h, 11.73 ± 1.00 μM at 48 h, and 2.92 ± 0.07 μM at 72 h). The same trend was inherent for its desulfated derivative **8a** but the activity was stronger against both cell lines in all time points (Table 8 and Table 9).

Evaluation of the selectivity index of the glycosides (Table 10) in relation to TNBC cells as compared with their toxicity against normal mammary epithelial cells showed the glycosides changed their activity over time. Conicospermiumoside A_3_-1 (**1**) showed maximal selectivity among all tested glycosides demonstrating the strongest cytotoxicity against the MDA-MB-231 cell line (SI 11.97) being simultaneously non-toxic against healthy cells at 48 h of exposition. Whereas, at 24 and 72 h its selectivity was significantly lessened. Conicospermiumoside A_7_-1 (**4**) (SI 8.47) and DS-okhotoside B_1_ (**8a**) (SI 9.06) more selectively acted against TNBC cells during 24 h, while okhotoside B_1_ (**8**) (SI 5.65) most effectively inhibited cancer cells during long exposition (72 h).

To assess the action of the glycosides on the formation and growth of tumor cell colonies directly related with the formation of metastases in the organism, a clonogenic assay was applied for the range of non-toxic concentrations of tested compounds against the MDA-MB-231 cell line (Figure 2). Additionally, colonies of MCF-10A were used to estimate this action of the glycosides in relation to healthy cells. It was found that the colony-formation ability of normal cells was much less susceptible to the glycosides action: the maximal inhibiting action was 40.2% of the control at maximum of used concentrations of 5 μM for okhotoside B_1_ (**8**) (Figure 2e). While even maximal used concentrations of the glycosides **4** and **8a**—2 μM and 0.5 μM—did not influence the colony formation and growth of MCF-10A cells. A dose-dependent effect of colony growth inhibition of MDA-MB-231 cells was observed for all tested compounds and almost complete stoppage was shown by conicospermiumosides A_3_-1 (**1**) (99.6% of inhibition out of control, Figure 2a) and A_3_-3 (**3**) (98.8% of inhibition out of control, Figure 2b) at maximal concentrations of 5 μM for both, and by DS-okhotoside A_1_-1 (**7a**) at concentration of 1 μM (97.2% of inhibition out of control, Figure 2d). Compound **3** also inhibited colony growth by 96.8% even at 2 μM concentration, and desulfated derivative **7a** suppress this process at a dose of 1 μM by 97.2% (Figure 2d).

The same series of glycosides in the concentration range below their EC_50_ to a different extent inhibited the migration of breast cancer MDA-MB-231 cells that was deduced from scratch analysis. In the control group of the untreated MDA-MB-231 cells, the scratch was overgrown within 24 h (Figure 3a). The greatest constraint on cell motility to 97% and 85% was demonstrated by okhotoside B_1_ (**8**) and DS-okhotoside A_1_-1 (**7a**) at concentrations of 5 μM and 1 μM, respectively (Figure 3b). The latter compound (**7a**) and conicospermiumoside A_3_-3 (**3**) significantly inhibited migration (by 61.2% and 63.7%, respectively, as compared with control) of MDA-MB-231 cells even at the doses of 0.5 μM and 2 μM, respectively.

### 2.3. Correlational Analysis and QSARs Model

The quantitative structure–activity relationships (QSARs) calculation was applied to analyze the correlations between the values of cytotoxic activity against human erythrocytes, the MCF-10A and MDA-MB-231 human cell lines, and structural peculiarities of the glycosides isolated from *C. conicospermium*, as well as to compare the obtained data with the observed patterns of SARs. Three-dimensional models of five new glycosides were built, protonated at pH 7.4, and subjected to energy minimization with MOE 2020.0901 CCG software Version 2020.09 [36] to select the predominant glycoside conformations for further analysis. For 38 conformations of 25 glycosides, a set of 2D and 3D descriptors (356 in total) describing physicochemical and geometric properties of molecules was calculated and analyzed using the QuaSAR-Descriptor tool of the MOE 2020.0901 CCG software [36]. In addition to the descriptors provided by the Calculate Descriptors tool of MOE software, the descriptors known to make significant impact to the activity of the glycosides (the presence/absence of 18(20)-lactone and the aglycone’s side chain, the presence/absence of substituents at C-16, C-22, C-23, the type of functionality at C-22, the position of the intranuclear double bond, the quantity of monosaccharides and carbohydrate chain branching, the type of the second and third sugar residues, and the number and positions of sulfate groups) were added to calculations.

A correlational analysis of the cytotoxic activity of the glycosides of *C. conicospermium* against human erythrocytes and MDA-MB-231 human cell lines revealed a good coincidence with the results previously obtained for glycosides isolated from the sea cucumber *C. djakonovi* [7,30]. To characterize the cytotoxic activity of the investigated glycosides against the non-tumor mammary epithelial cell line MCF-10A from the structural point, the correlation analysis was performed. An analysis of the principal components (PCA) resulted in the division of glycosides into two groups (Figure 4), confirming the correct descriptor’s choice. The linear QSARs model for the cytotoxic activity of 25 investigated glycosides against MCF-10A cells was also constructed using the QuaSAR-Model tool of the MOE 2020.0901 CCG software [36], as described earlier [30]. The model fits well with the experimental data on the cytotoxic activities of glycosides with a correlation coefficient r^2^ = 0.96501 and RMSE = 0.05670 (Appendix A). The model was cross-validated with r^2^_cros_ = 0.83214 and RMSE_cros_ = 0.21643. The QSARs model includes 64 terms, 48 from those that made an important contribution.

The QSARs model of the cytotoxic effects of glycosides against MCF-10A cells showed the significant positive impact of the following structural elements as follows: presence of the aglycone side chain, 18(20)-lactone, 16-OAc and 23-keto-group, 7(8)-position of the intranuclear double bond, and the absence of sulfate group at the fourth residue in carbohydrate chain. Whereas, the branching of the carbohydrate chain, an increase of the sulfation degree, especially the appearance of the sulfate group at the terminal (fourth) residue, the 9(11)-position of the double bond, the absence of a side chain, and the presence of the 25-OH-group, all made a negative contribution to the activity. Noticeably, the majority of descriptors significant for cytotoxicity against MCF-10A cells were characterized by their similar contributions (positive or negative) to hemolysis and cytotoxicity against the MDA-MB-231 cell line, while some have changed the impact considerably (Appendix A). These calculations were corroborated by different selectivity indexes (Table 10) demonstrated by glycosides in relation to healthy and TNBC cells.

Thus, it was shown that the presence of glucose as the second residue slightly negatively correlated with activity against the MCF-10A cells but made a substantial positive contribution to toxicity in relation to the MDA-MB-231 cell line. The degree of sulfation, especially the presence of sulfate groups at C-6 of the third and fourth residues in the sugar chain was shown to reduce toxicity (three or fivefold) against MCF-10A cells, as opposed to toxicity against erythrocytes and MDA-MB-231 cells. The 16-*O*-acetylation was also shown to be significantly positively correlated with the cytotoxicity against MDA-MB-231 cells, whereas the contribution to cytotoxicity against MCF-10A cells and hemolytic activity was not so strong. The structure of the side chain influenced activity in different ways. The appearance of a keto-group at C-22 significantly increased the hemolytic activity, and almost did not affect the cytotoxicity toward MCF-10A and MDA-MB-231 cells; the same tendency was observed for the 24(25)-double bond, while the presence of the 22-hydroxy group had no impact to hemolysis, positively influenced to anti-MCF-10A activity and negatively to activity against MDA-MB-231. Oppositely, the 25(26)-double bond significantly promoted cytotoxicity against MDA-MB-231 cells, slightly influenced to cytotoxicity against normal cells, but negatively affected hemolytic activity.

Such variability of impacts was reflected in the values of selectivity indexes (SI) (Table 10). Thus, the analysis of the descriptor’s correlations (Appendix A) evaluated for conicospermiumoside A_3_-1 (**1**) demonstrating the highest SI to tumor cells showed opposite impacts of the second quinovose and the 22-OH-group to the activity against MCF-10A and MDA-MB-231 cells, and varying degrees of impact of the sulfate group at C-6 Glc3. For trisulfated conicospermiumoside A_7_-1 (**4**), which was an order of magnitude less active against MCF-10A than to MDA-MB-231 cells, the sustainable differences in correlations were found for sulfate group at C-6 MeGlc4 and the 9(11)-double bond; the 22-oxo-group also slightly differently influenced the activity against normal and cancer cells. A correlational analysis of descriptors, characteristic of okhotoside B_1_ (**8**) and desulfated derivative **8a** selectively active against tumor cells at different time points, showed a strong positive impact to the anticancer activity of the second glucose residue, the terminal double bond, and the 16-*O*-acetic group, while the availability of the sulfate group strongly reduced the cytotoxicity of **8** against MCF-10A cells.

## 3. Discussion

### 3.1. Glycosides Structural Features and Biogenetic Analysis

A total of seventeen compounds were isolated from the glycosidic fraction of *C. conicospermium*. Five of them turned out to be new glycosides having non-holostane aglycones differing by side chain structures, as well as by the quantity of sulfate groups (two or three) in pentasaccharide chains. From the five compounds found earlier as result of the first investigation of glycosidic fraction of *C. conicospermium* [9], three glycosides (**12**, **16**, **17**) were also identified this time. From the whole series, only conicospermiumoside A_7_-1 (**4**) and isokoreoside A (**17**) have the 9(11)-double bond in the aglycones, while the other 15 glycosides contained the 7(8)-intranuclear double bond. This situation is similar to that observed for *C. djakonovi*, where isokoreoside A (**17**) was a single compound from the series of 20 glycosides having a 9(11)-double bond. By contrast, *C. frondosa* contained four pairs of glycosides isomeric by the position of a double bond in the lanostane nuclei [16]. These data indicate that two oxidosqualene cyclases (OSCs) (parkeol syntase and 9βH-lanosta-7,24-diene-3β-ol syntase), differing by their functional activity, are inherent for sea cucumbers of the *Cucumaria* genus.

Nine compounds isolated from *C. conicospermium* were characterized by non-holostane aglycones without lactone, including two hexa-nor-lanostane aglycones in koreoside A (**16**) and isokoreoside A (**17**). A comparison of the side chain structures of some lanostane aglycones differing by substituents at C-22 led to the construction of a biogenetic scheme which ended by the formation of hexa-nor-lanostane aglycones (in **16**, **17**) (Figure 5). A similar set of glycosides was found in *C. frondosa* [16], but now the blanks are filled with those aglycones having the 22-keto-groups are similar to the direct precursors of vertebrates’ sex steroid hormones without side chains.

Eight known glycosides, including djakonoviosides A (**6**) and B_2_ (**13**), okhotosides A_1_-1 (**7**), B_1_ (**8**), B_2_ (**9**), and A_2_-1 (**11**), frondoside A (**10**), and cucumarioside A_2_-5 (**12**), contained holostane-type aglycones. Interestingly, non-holostane aglycones are the parts of disulfated and trisulfated pentaosides, while holostane aglycones comprise monosulfated tetra- and pentaosides, excepting the disulfated tetraoside okhotoside B_2_ (**9**). Considering that formation of the 18(20)-lactone ring in holostane-type aglycones is a downstream process in relation to biosynthesis of non-holostane-type derivatives, and that sulfation of carbohydrate chains during biosynthesis is one of the terminal stages, it becomes obvious that a sort of functionally-structural division is realized in the process of biosynthesis of glycosides in *C. conicospermium*. A part of monosulfated tetrasaccharide lanostane precursors undergo subsequent transformation of aglycone parts (C-18 oxidation followed by intramolecular dehydration and cyclization) resulting in holostane aglycone formation; whereas the other part of lanostane derivatives is exposed to subsequent glycosylation and sulfation of their carbohydrate chains, while the stage of C-18 oxidation is missed, lactone is not formed, but oxidative transformations of the side chains by C-22 and further the 20(22)-bond cleavage with elimination of the side chain portion take place. It is noteworthy that holostane aglycones of the glycosides of *C. conicospermium* are all acetylated by C-16 (with exception of djakonovioside B_2_ (**13**) whose 16-hydroxylated precursor undergoes intramolecular cyclization followed by hemiketal-fragment formation [7]) and contains a non-oxidized side chain or is oxidized at C-23 only. This observation additionally confirmed the separation of biosynthetic pathways of holostane and lanostane derivatives. A similar structural separation of the glycosides consisting in the certain combinations of aglycone types and oligosaccharide chain types are also characteristic for *C. frondosa* [16].

Noticeably, koreoside A (**16**)—a trisulfated pentaoside with a shortened side chain—was present in a majority of the chemically studied *Cucumaria* species: *C. koraiensis koreaensis* [35], *C. conicospermium*, *C. frondosa*, *C. okhotensis*, *C. djakonovi*, and *C. fallax* [37]. Hence, the composition of less polar constituents of glycosidic sums of different *Cucumaria* species is more variable, than that of trisulfated glycosides. Moreover, in *C. conicospermium* the quantitative content of polar trisulfated glycosides was much higher than that of lesser polar fractions. The necessity of availability of the glycoside with very specific aglycone structure, structurally similar to sex hormones, as in koreoside A (**16**), and known to demonstrate low membranolytic activity indicates its responsibility for specific biological functions (chemical signaling regulating the reproductive process in the population) other than the defensive role inherent for the majority of sea cucumber glycosides.

Thus, the glycosides of *C. conicospermium* were structurally similar with those of *C. frondosa*, including four glycosides common for both species, with *C. djakonovi* sharing eight common glycosides, and with *C. okhotensis* sharing the other set of seven common glycosides. The obtained chemical data resembled the biochemical and phylogenetic closeness of the species belonging to the *Cucumaria* genus. Such chemical “overlapping” hampers the use of triterpene glycosides as chemotaxonomic markers for sea cucumbers of the genus *Cucumaria*. Despite some successive examples of using the glycosides to resolve systematic and phylogenetic difficulties of the sea cucumber species identification [4,6,36], the application of a comprehensive approach to the systematic identification of specimens supposed to belong to the genus *Cucumaria* is urgently needed.

The studied sea cucumbers were collected near Onekotan Island during 36th expedition of the research vessel, “Academic Oparin”, and were identified by an analysis of morphologic features and ossicle shape (Figure 6). The very specific type of ossicles, elongated perforated plates with holes evenly distributed on the surface, tapered at one end and ending in a sharp spike as well as analysis of geographic distribution [38], allowed us to identify the samples as *Cucumaria conicospermium*. The set of isolated glycosides fits in well with the data obtained earlier for the glycosidic composition of the species [9]. In addition, the set of glycosides inherent to *C. conicospermium* was broadened with five new compounds and nine known compounds.

Eight types of carbohydrate chains comprised the glycosides of *C. conicospermium*. Among them are monosulphated tetraosides (**6**–**8**) differing by the second sugar unit (quinovose or glucose), disulfated tetraoside **9**, three types of monosulfated pentaosides having quinovose or glucose in the second position of the sugar chain and xylose or glucose as the third monosaccharide residue (**10**–**13**). Disulfated (**1**–**3**) and trisulfated (**4**, **5**, **14**–**17**) glycosides were pentaosides sharing the same monosaccharide composition. The set of carbohydrate chains of *C. conicospermium* consisted of known earlier parts of glycosides of sea cucumbers of the *Cucumaria* genus. However, the composition of sugar chains of *C. conicospermium* was different from the sugar chains found in *C. japonica* [4], *C. frondosa* [5,13,14,15,16], and *C. djakonovi* [7,8] and was most similar to the carbohydrate chains set of *C. okhotensis* [6,17].

### 3.2. Observed and Calculated Structure–Activity Relationships (SARs and QSARs) Comparison

The analysis of hemolytic activity of new glycosides (**1–5**), okhotoside A_1_-1 (**7**) (as positive control) and known compounds, that have not been tested earlier against breast cancer cell lines: okhotoside B_1_ (**8**) and B_2_ (**9**), desulfated derivatives **7a** and **8a**, frondoside A (**10**), and frondoside A_7_-4 (**14**) demonstrated ten compounds from the series were strongly active. The highest hemolytic activities were shown by okhotoside A_1_-1 (**7**) and its desulfated derivative **7a** confirming known regularity that glycosides possessing holostane aglycone and a linear tetrasaccharide chain (bearing or not one sulfate group) with quinovose residue in the second position are highly membranolytic [39]. Okhotosides B_1_ (**8**) and B_2_ (**9**), differing from each other by sulfate group quantity and characterized by glucose as the second sugar in their tetrasaccharide linear chains, were almost equally active but less active than quinovose-containing glycoside **7**. Desulfation of the monosulfated compound **8** led to an increase of the hemolytic properties. Other strongly hemolytic compounds were conicospermiumosides A_3_-3 (**3**) and A_7_-1 (**4**), having the same lanostane aglycones with the 22-oxo,24(25)-en fragment in the side chains and differing by the position of the double bond in the polycyclic nuclei, and by the quantity of sulfate groups. Noticeably, djakonovioside F_1_ (**15**), having the same aglycone as **3** and studied earlier, was also highly active [8]. The quantity of the sulfate groups does not really matter in this case. Disulfated conicospermiumoside A_3_-1 (**1**) was three times more active than trisulfated frondoside A_7_-4 (**14**) with the same aglycones having the 22-hydroxy-group. Acetylation of the 22-OH group, as in conicospermiumoside A_3_-2 (**2**), led to an almost 10-fold decrease of activity. The less active compound in relation to erythrocytes was conicospermiumoside A_7_-2 (**5**) having two hydroxyl groups in the side chain confirming the knowledge about the negative influence this functionality to the membranolytic properties of the glycosides [39].

The results of applying a QSARs analysis, linking the structures of glycosides and the activity against different cell lines, allowed us to detect the structural features of the glycosides, which correlate differently, sometimes even in opposite ways, with their action toward cells (erythrocytes, MCF-10A, and TNBC MDA-MB-231 cells). This observation allows us to assume that, when interacting with diverse cell types, glycosides target different membrane components. Apparently, when acting on erythrocytes, their objects are the lipid components of the membrane while, when interacting with human mammary cells (tumor or non-tumor), such targets may be receptors embedded in the membrane. Considering the relationships between the glycoside structures and cytotoxicity in relation to the non-tumor cell line MCF-10A and the tumor MDA-MB-231 cell line, most likely the glycosides act on distinct receptors (or receptors with different expression level) inherent for these cell types.

### 3.3. Biological Activity of the Glycosides Against Breast Cancer Cells

The sensitivity of MDA-MB-231 cells to the cytotoxic action of the glycosides, as well as relative sustainability of the MCF-7 cell line, was demonstrated with a tested series of the glycosides which correlated well with earlier observations [7,8]. The T-47D cell occupied an intermediate position regarding to sensitivity to glycoside action between other tested breast cancer cells. The series of six glycosides was chosen for a more detailed study of some aspects of their anticancer action against MDA-MB-231 cells. The selective action of the compounds **4** and **8** on TNBC cells without significant toxicity against MCF-10A cells at 24-h experiment was observed. It was noticeable that the selectivity of the compounds was changed with an increasing duration of the experiment, and maximal selectivity to cancer cells was demonstrated by glycoside **1** at 48 h of exposition. Additionally, glycosides in the series differed by their anti-proliferative properties. All the compounds retained the activity against MDA-MB-231 cells after 48 and 72 h of exposition; however, part of them became more toxic toward normal MCF-10A cells. Conicospermiumosides A_3_-1 (**1**), A_7_-1 (**4**), and okhotoside B_1_ (**8**) look most favorable in this respect, being slightly cytotoxic against normal cells in opposition to cancer cells.

All the substances from the series were able to inhibit colony formation and growth, as well as migration of TNBC cells to a different extent. Conicospermiumoside A_3_-1 (**1**) most effectively suppressed the colony-formation ability of cancer cells; similar properties were demonstrated by conicospermiumoside A_3_-3 (**3**) and the desulfated derivative **7a**. In relation to the cell motility inhibiting capacity, okhotoside B_1_ (**8**) and DS-okhotoside A_1_-1 (**7a**) were the most active, but conicospermiumoside A_3_-3 (**3**) was also among the promising compounds. The studied glycosides had different effects on some aspects of cancer cell functioning, indicating some of them should be further investigated as potential anticancer drugs.

## 4. Materials and Methods

### 4.1. General Experimental Procedures

PerkinElmer 343 Polarimeter (PerkinElmer, Waltham, MA, USA) was used for specific rotation measuring; NMR spectra were registered on an Avance III 700 Bruker FT-NMR spectrometer (Bruker BioSpin GmbH, Rheinstetten, Germany) (700.13/176.04 MHz (^1^H/^13^C, 30 °C, δ_C_ 148.9 resonance of C_5_D_5_N for ^13^C and δ_H_ 7.21 resonance of C_5_D_5_N for ^1^H used as the references, BBO probe)); the ESI MS (positive and negative ion modes) spectra were obtained on an Agilent 6510 Q-TOF apparatus (Agilent Technology, Santa Clara, CA, USA), with a sample concentration of 0.01 mg/mL; HPLC was conducted on an Agilent 1260 Infinity II equipped with a differential refractometer (Agilent Technology, Santa Clara, CA, USA); columns were used: Phenomenex Synergi Fusion-RP (10 × 250 mm) (Phenomenex, Torrance, CA, USA) and Supelco Discovery HS F5-5 (10 × 250 mm, 5 µm) (Supelco, Bellefonte, PA, USA) (flow rate of 1.5 mL/min).

### 4.2. Animals and Cells

The specimens of sea cucumber *C. conicospermium* Levin et Stepanov (family Cucumariidae; order Dendrochirotida) were collected near Onekotan Island (on the Sea of Okhotsk side) during the 36th expedition of the research vessel, “Academic Oparin”, by dragging from a depth of 91 m in August 2008. The taxonomic identification of the animals was performed by Dautov S.Sh. Voucher specimen is kept in the A.V. Zhirmunsky National Scientific Center of Marine Biology, Far Eastern Branch, Russian Academy of Sciences.

Human erythrocytes were purchased from the Station of Blood Transfusion, Vladivostok. Human mammary epithelial cell line MCF-10A CRL-10317, human breast cancer cell lines T-47D HTB-133, MCF-7 HTB-22, and MDA-MB-231 CRM-HTB-26 were received from ATCC (Manassas, VA, USA). Culturing conditions: medium of RPMI-1640 with 1% penicillin/streptomycin (Biolot, St. Petersburg, Russia) and 10% fetal bovine serum (FBS) (Biolot, St. Petersburg, Russia) for the T-47D cell line; Minimum Essential Medium (MEM) with 1% penicillin/streptomycin sulfate (Biolot, St. Petersburg, Russia) and FBS (Biolot, St. Petersburg, Russia) to a final concentration of 10% for MCF-7 and MDA-MB-231 cells; DMEM/F12 medium with 10% FBS, 20 ng/mL EGF, 0.5 mg/mL hydrocortisone, 100 ng/mL cholera toxin, 10 μg/mL insulin, and 1% penicillin/streptomycin (BioinnLabs, Rostov-on-Don, Russia) for MCF-10A cell line.

### 4.3. Extraction and Isolation

The frozen sea cucumbers (23.5 kg) were minced and extracted twice with refluxing 70% EtOH. The obtained extract was evaporated, dissolved in H_2_O, filtered, and subjected to hydrophobic chromatography on a Polychrom-1 column (powdered Teflon, Biolar, Latvia) for the elimination of inorganic salts and impurities. A crude glycoside fraction (3800 mg) was obtained as a result of elution with 50% ethanol. Subsequently, it was separated by column chromatography on Si gel with the staggered gradient of the systems of eluents of CHCl_3_/EtOH/H_2_O in ratios of 100:50:4, 100:75:10, 100:100:17, 100:125:25, and 100:150:40. The fractions 1–5 were obtained and submitted to HPLC on reversed-phase semipreparative columns, Phenomenex Synergi Fusion RP (10 × 250 mm), and Supelco Discovery HS F5-5 (10 × 250 mm). HPLC of fraction 1 on a reversed-phase column Synergi Fusion RP (10 × 250 mm) with MeOH/H_2_O/NH_4_OAc (1M water solution) in a ratio of (70/28/2) as mobile phase gave okhotoside A_1_-1 (**7**) (5.4 mg) and two other subfractions. One of them was repeatedly subjected to HPLC with the same solvent system in a ratio of (72/36/2) that resulted in isolation of 2.6 mg of frondoside A (**10**). The re-chromatography of another subfraction with CH_3_CN/H_2_O/NH_4_OAc (1M water solution) (40/58/2) as a mobile phase resulted in the isolation of djakonovioside A (**6**) (4.6 mg). Fraction 2, chromatographed with MeOH/H_2_O/NH_4_OAc (1M water solution) in a ratio of (70/28/2) as a mobile phase, gave individual cucumarioside A_2_-5 (**12**) (7.5 mg), as well as two subfractions 2.1 and 2.3. The re-chromatography of 2.3 with the same solvent system in a ratio of (72/36/2) led to pure okhotoside B_1_ (**8**) (3.4 mg). HPLC of subfraction 2.1 on a Discovery HS F5-5 (10 × 250 mm) column with CH_3_CN/H_2_O/NH_4_OAc (1M water solution) (33/64/3) as a mobile phase resulted in the isolation of okhotoside A_2_-1 (**11**) (8.9 mg) and djakonovioside B_2_ (**13**) (2.3 mg). HPLC of fractions 3 and 4 was conducted on a Supelco Discovery HS F5-5 (10 × 250 mm) column. Fraction 3 mainly consisted of okhotoside B_2_ (**9**) (3.2 mg) isolated as a result of chromatography with CH_3_CN/H_2_O/NH_4_OAc (1M water solution) (36/62/2) as a mobile phase. Fraction 4 was chromatographed with CH_3_CN/H_2_O/NH_4_OAc (1M water solution) (35/63/3) as a mobile phase gave some subfractions. One of which, after repeated separation procedure with the same mobile phase but in a ratio (33/64/3), gave conicospermiumosides A_3_-2 (**2**) (2.4 mg), and A_3_-3 (**3**) (2.9 mg), another subfraction gave conicospermiumoside A_3_-1 (**1**) (4.1 mg) after its HPLC with the same solvents in a ratio (31/66/3). The most polar fraction 5 demonstrated better separation on the column Synergi Fusion RP (10 × 250 mm) with the solvent system MeOH/H_2_O/NH_4_OAc (1M water solution) in a ratio of (67/30/3) as a mobile phase to give 5 subfractions (5.1–5.5). Subfraction 5.5 allowed us to isolate conicospermiumoside A_7_-1 (**4**) (4.1 mg) as a result of re-chromatography with the same solvents in a ratio of (70/27/3); subfraction 5.4 mainly contained djakonovioside F_1_ (**15**) (10.2 mg) isolated in the same conditions as **4**. Frondoside A_7_-4 (**14**) (2.9 mg) was isolated from the subfraction 5.3 after HPLC with MeOH/H_2_O/NH_4_OAc (1M water solution) in a ratio of (68/29/3). HPLC of subfraction 5.2 was with a mobile phase in another ratio (61/35/4) led to separation of koreoside A (**16**) (13.4 mg), and isokoreoside A (**17**) (2.5 mg). Conicospermiumoside A_7_-2 (**5**) (1.6 mg) was obtained from subfraction 5.1 after its separation with the same solvents in a ratio (54/41/5).

#### 4.3.1. Conicospermiumoside A_3_-1 (**1**)

Colorless powder; [α]_D_^20^−63° (*c* 0.1, H_2_O), mp 180–182 °C. NMR: Table 1 and Table 2, Appendix A. (−)HR-ESI-MS *m*/*z*: 1387.5269 (calc. 1387.5280) [M_2Na_–Na]^−^, 682.2704 (calc. 682.2694) [M_2Na_–2Na]^2−^; (−)ESI-MS/MS *m/z*: 1267.5 [M_2Na_ − Na − NaHSO_4_]^−^, 1255.5 [M_2Na_ − Na − Xyl (C_5_H_8_O_4_)]^−^, 1153.5 [M_2Na_ − Na − NaSO_3_ − Xyl (C_5_H_8_O_4_) + H]^−^, 815.4 [M_2Na_ − Na − Agl (C_30_H_49_O_2_) − Xyl (C_5_H_8_O_4_) + H]^−^; (+)ESI-MS/MS *m*/*z*: 1433.5 [M_2Na_ + Na]^+^, 1331.6 [M_2Na_ + Na − NaSO_3_]^+^, 1301.6 [M_2Na_ + Na − Xyl (C_5_H_8_O_4_)]^+^, 1241.5 [M_2Na_ + Na − MeGlc (C_7_H_13_O_6_)+ H]^+^, 1097.6 [M_2Na_ + Na − Xyl (C_5_H_8_O_4_) − 2NaSO_3_ + H]^+^, 1023.5 [M_2Na_ + Na − Xyl (C_5_H_8_O_4_) − NaSO_3_ − MeGlc (C_7_H_12_O_5_)+ H]^+^, 873.2 [M_2Na_ + Na − Agl (C_30_H_49_O_3_) − NaSO_3_]^+^, 741.1 [M_2Na_ + Na − Agl (C_30_H_49_O_3_) − XylSO_3_ (C_5_H_7_O_3_SNa) + H]^+^, 565.1 [M_2Na_ + Na − Agl (C_30_H_49_O_3_) − XylSO_3_ (C_50_H_7_O_3_SNa) − MeGlc (C_7_H_13_O_5_)+ H]^+^.

#### 4.3.2. Conicospermiumoside A_3_-2 (**2**)

Colorless powder; [α]_D_^20^−78° (*c* 0.1, H_2_O), mp 201–203 °C. NMR: Table 3 and Appendix A. (−)HR-ESI-MS *m*/*z*: 1429.5381 (calc. 1429.5386) [M_2Na_–Na]^−^, 703.2761 (calc. 703.2747) [M_2Na_ − 2Na]^2−^; (−)ESI-MS/MS *m/z*: 1369.5 [M_2Na_ − Na − CH_3_COOH]^−^, 1249.5 [M_2Na_ − Na − NaHSO_4_ − CH_3_COOH]^−^; (+)ESI-MS/MS *m*/*z*: 1415.5 [M_2Na_ + Na − CH_3_COOH]^+^, 1355.5 [M_2Na_ + Na − NaHSO_4_]^+^, 1295.5 [M_2Na_ + Na − NaHSO_4_ − CH_3_COOH]^+^, 1223.5 [M_2Na_ + Na − NaHSO_4_ − Xyl (C_5_H_8_O_4_)]^+^, 991.7 [M_2Na_ + Na − Agl (C_32_H_51_O_3_)]^+^, 873.2 [M_2Na_ + Na − Agl (C_32_H_51_O_3_) − NaSO_4_]^+^, 741.1 [M_2Na_ + Na − Agl (C_32_H_51_O_3_)– XylSO_3_ (C_5_H_7_O_8_SNa)]^+^.

#### 4.3.3. Conicospermiumoside A_3_-3 (**3**)

Colorless powder; [α]_D_^20^−75° (*c* 0.1, H_2_O), mp 191–193 °C. NMR: Table 4 and Appendix A. (−)HR-ESI-MS *m*/*z*: 1385.5111 (calc. 1385.5124) [M_2Na_ − Na]^−^; 681.2622 (calc. 681.2616) [M_2Na_ − 2Na]^2−^; 1283.5719 (calc. 1283.5736) [M_2Na_ − Na − SO_3_Na + H]^−^; 1203.6159 (calc. 1203.6168) [M_2Na_ − Na − 2SO_3_Na + 2H]^−^; (−)ESI-MS/MS *m*/*z*: 1151.5 [M_2Na_ − Na − SO_3_Na − Xyl (C_5_H_8_O_4_) + H]^−^, 1027.5 [M_2Na_ − Na − 2SO_3_Na − MeGlc + H]^−^, 865.5 [M_2Na_ − Na − 2SO_3_Na − MeGlc − Glc + H]^−^, 733.5 [M_2Na_ − Na − 2SO_3_Na − MeGlc (C_7_H_13_O_5_) − Glc (C_6_H_10_O_5_) − Xyl (C_5_H_8_O_4_) + H]^−^, 695.2 [M_2Na_ − Na − SO_3_Na − Xyl (C_5_H_8_O_4_) − Agl (C_30_H_47_O_3_)]^−^.

#### 4.3.4. Conicospermiumoside A_7_-1 (**4**)

Colorless powder; [α]_D_^20^−68° (*c* 0.1, H_2_O), mp 172–174 °C. NMR: Table 5 and Table 6, Appendix A. (−)HR-ESI-MS *m*/*z*: 732.2305 (calc. 732.2310) [M_3Na_− 2Na]^2−^, 480.4910 (calc. 480.4909) [M_3Na_ − 3Na]^3−^; (+)HR-ESI-MS *m*/*z*: 1533.4285 (calc. 1533.4296) [M_3Na_ + Na]^+^; (−)ESI-MS/MS *m*/*z*: 681.3 [M_3Na_ − 2Na − SO_3_Na + H]^2−^, 666.2 [M_3Na_ − 2Na − Xyl]^2−^, 615.2 [M_3Na_ − 2Na − Xyl − SO_3_Na + H]^2−^; (+)ESI-MS/MS *m*/*z* 1431.5 [M_3Na_ + Na − SO_3_Na + H]^+^, 1413.5 [M_3Na_ + Na − NaHSO_4_]^+^, 1329.5 [M_3Na_ + Na − 2SO_3_Na + 2H]^+^, 1255.4 [M_3Na_ + Na − MeGlcSO_3_ (C_7_H_12_O_8_SNa) + H]^+^, 975.1 [M_3Na_ + Na − SO_3_Na − Agl (C_30_H_47_O_3_)]^+^, 843.1 [M_3Na_ + Na − SO_3_Na − Agl (C_30_H_47_O_3_) − Xyl (C_5_H_8_O_4_)]^+^.

#### 4.3.5. Conicospermiumoside A_7_-2 (**5**)

Colorless powder; [α]_D_^20^−74° (*c* 0.1, H_2_O), mp 184–186 °C. NMR: Table 7 and Appendix A. (−)HR-ESI-MS *m*/*z*: 741.2344 (calc. 741.2362) [M_3Na_ − 2Na]^2−^, 486.4942 (calc. 486.4944) [M_3Na_ − 3Na]^3−^; (−)ESI-MS/MS *m*/*z*: 847.4 [M_3Na_ − Na −MeGlcSO_3_ − GlcSO_3_ − Xyl + H]^−^, 797.1 [M_3Na_ − Na − Agl − XylSO_3_ − H]^−^, 519.0 [M_3Na_ − Na − Agl − XylSO_3_ − MeGlcSO_3_ − H]^−^, 690.3 [M_3Na_ − 2Na − NaSO_3_]^2−^, 683.2 [M_3Na_ − 2Na − C_6_H_11_O_2_ − H]^2−^.

### 4.4. Cytotoxic Activity (MTT Assay)

The concentrations of tested glycosides were 0.1–50 µM; positive controls were cisplatin and okhotoside A_1_-1 (**7**) [7]. Methodology: to each well of 96-well plates, the cell suspension (180 µL) with solution (20 µL) of tested glycoside in a certain concentration was placed (MCF-10A, MCF-7, T-47D, and MDA-MB-231—7 × 103 cells per well) and incubated in the atmosphere with 5% CO_2_ at 37 °C for 24 h. Then, the solutions of tested compounds with medium were replaced by 100 µL of fresh medium, and 10 µL of 3-(4,5-dimethylthiazol-2-yl)-2,5-diphenyltetrazolium bromide (MTT) (PanReac, AppliChem, Darmstadt, Germany) stock solution (5 mg/mL) was added to each well and incubated for 4 h, followed by the addition of 100 µL of SDS-HCl solution (1 g SDS/10 mL d-H2O/17 µL 6 N HCl) and further incubated for 18 h. Multiskan FC microplate photometer (Thermo Fisher Scientific, Waltham, MA, USA) was used to measure the absorbance of the converted dye formazan at 570 nm. The concentration caused the 50% cell metabolic activity inhibition (IC50) expresses the cytotoxic activity of each glycoside. The experiments were conducted in triplicate, *p* ≤ 0.05.

### 4.5. Hemolytic Activity

Human blood (B(III) Rh+) was used to obtain erythrocytes by centrifuging 450× *g* three times for 5 min with phosphate-buffered saline (PBS) (pH 7.4) at 4 °C on centrifuge LABOFUGE 400R (Heraeus, Hanau, Germany). Ice-cold PBS (pH 7.4) was used for resuspension of erythrocytes residue to a final optical density of 1.5 at 700 nm, which was kept on ice. Then, 10 µL of tested compound solution or control (okhotoside A_1_-1 (**7**)) were added to 90 µL of erythrocyte suspension in V-bottom 96-well plates and exposed for 1 h at 37 °C. Next, centrifugation at 900× *g* for 10 min on a laboratory centrifuge LMC-3000 (Biosan, Riga, Latvia), led to separation of layers, and 80 µL of supernatant was carefully decanted and transferred into new flat-plate for each. The values of erythrocyte lysis were measured on microplate photometer Multiskan FC (Thermo Fisher Scientific, Waltham, MA, USA) at λ = 570 nm as hemoglobin concentration in supernatant. The effective dose, causing lysis of 50% erythrocytes (ED50), was calculated with SigmaPlot 14.0 software. All the experiments were carried out in triple repetitions, *p* ≤ 0.05.

### 4.6. Colony Formation Assay

The influence of glycosides on colony formation by MCF-10A or MDA-MB-231 cells was tested by the clonogenic assay [40]. Cell density: 0.3 × 10^2^/mL for MDA-MB-231 and for MCF-10A cells per well; incubation conditions: 10–12 days, 37 °C, atmosphere with 5% CO_2_; obtained: visible to eye colonies (at least 50 cells per colony); fixation with methanol (25 min); staining with 0.5% solution of crystal violet (25 min); washing and air-drying of plates. The counting of grown colonies was carried out using a BIO-PRINT-Cx4 Edge-Fixed Pad-Container (Vilber, Collégien, France) using Bio-Vision Software user and service manual-v18.01 (Vilber, Collégien, France). The results are presented as colony inhibition in comparison with the control.

### 4.7. Wound Scratch Migration Assay

Attached to special migration plate plastic bottom MDA-MB-231 cells were separated by a silicone insert (Culture-insert 2 Well 24, ibiTreat). After removing an insert, the gap between the cells was 500 ± 50 μm. Cell debris and floating cells were deleted, and the fresh culture medium was added. Then, cells were treated with various concentrations of glycosides or culture medium only (vehicle control) and left for 24 hrs. Cell migration into the wound area was observed under a fluorescence microscope (MIB-2-FL, LOMO, Russia) with objective 10× magnification.

### 4.8. Building a QSARs Model

A QSARs model for the set of 25 glycosides was built using QuaSAR-Descriptor and QuaSAR-Model tools of MOE 2020.0901 software [36]. The procedure involved the following steps: a charge calculation and structure optimization, glycosides conformational search, descriptors calculation, correlational analysis, principal component analysis (PCA), removing the descriptors collinear with another descriptor (unnecessary descriptors), building a QSARs model and the model’s cross-validation, removing the descriptors not contributing to the model, and model checking by making a graph showing the correlation between the model-predicted value and the experimental activity value expressed as pED_50_ or pIC_50_.

## 5. Conclusions

As a result of investigation of the glycosidic composition of the sea cucumber *Cucumaria conicospermium*, five new conicospermiumosides and twelve known glycosides, found earlier in other representatives of the *Cucumaria* genus, were isolated. Eleven different aglycones and eight types of carbohydrate chains composed the glycosides. All novel glycosides had lanostane-type aglycones without lactone and differed mainly by the side chain structures. Two of the glycosides from the isolated series contained the 9(11)-double bond, while the others were characterized by the 7(8)-double bond.

The biogenetic row of aglycones leading to the formation of hexa-nor-lanostane derivatives probably performing regulatory function in the holothurians population was proposed. A functionally-structural division of the glycosides, realized as a result of the separation of biosynthetic pathways of holostane and nor-lanostane derivatives, was deduced.

The cytotoxic action against three human breast cancer cell lines and non-tumor mammary cells was tested, along with the hemolytic activity of compounds **1**–**5**, as well as the seven known glycosides isolated from *C. conicospermium*. Some of the glycosides demonstrated promising anti-breast-cancer effects, such as strong inhibition of cells’ motility and colony-forming ability, in rather low concentrations. Noticeably, some compounds were selectively active against TNBC cells in contrast to being non-cytotoxic against normal mammary epithelial cells.

The QSARs calculation linking the structures of glycosides and the activity against different cell lines showed some structural features correlated differently, sometimes even in opposite ways, with their action toward cells (erythrocytes, MCF-10A, and TNBC MDA-MB-231 cells). This observation indicated that glycosides obviously target different membrane components.

## Figures and Tables

**Figure 1 marinedrugs-22-00560-f001:**
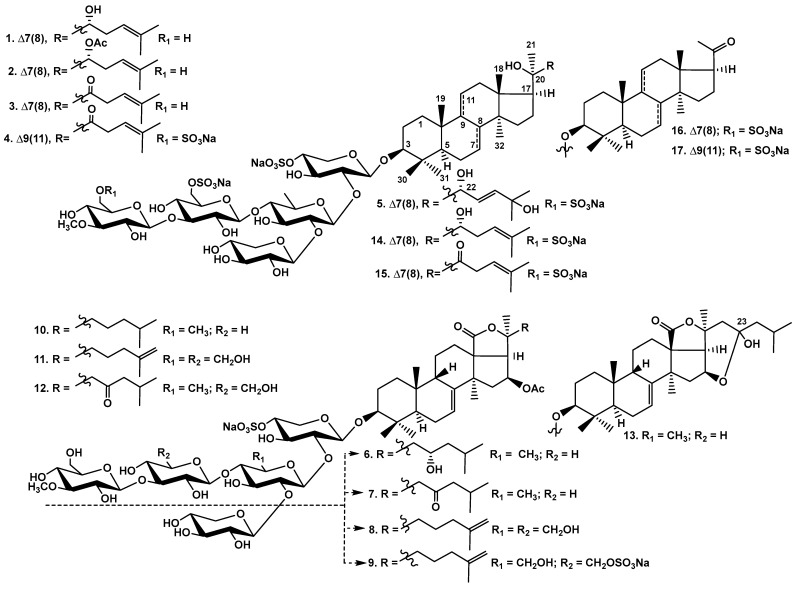
Chemical structures of glycosides of *Cucumaria conicospermium*. New glycosides: **1**—conicospermiumoside A_3_-1; **2**—conicospermiumoside A_3_-2; **3**—conicospermiumoside A_3_-3; **4**—conicospermiumoside A_7_-1; **5**—conicospermiumoside A_7_-2. Known compounds: **6**—djakonovioside A; **7**—okhotoside A_1_-1; **8**—okhotoside B_1_; **9**—okhotoside B_2_; **10**—frondoside A; **11**—okhotoside A_2_-1; **12**—cucumarioside A_2_-5; **13**—djakonovioside B_2_; **14**—frondoside A_7_-4; **15**—djakonovioside F_1_; **16**—koreoside A; **17**—isokoreoside A.

**Figure 2 marinedrugs-22-00560-f002:**
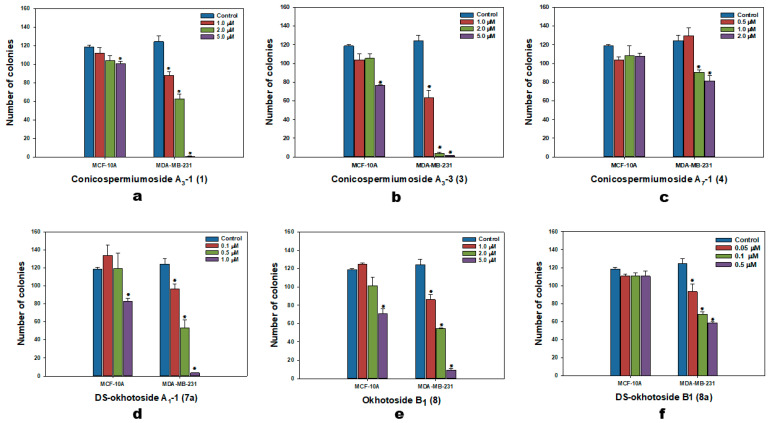
The number of MCF-10A and MDA-MB-231 cell colonies under treatment with different concentrations of the glycosides: (**a**)—conicospermiumoside A_3_-1 (**1**) at concentrations 1, 2, and 5 μM, (**b**)—conicospermiumoside A_3_-3 (**3**) at concentrations 1, 2, and 5 μM, (**c**)—conicospermiumoside A_7_-1 (**4**) at concentrations 0.5, 1, and 2 μM, (**d**)—DS-okhotoside A_1_-1 (**7a**) at concentrations 0.1, 0.5, and 1 μM, (**e**)—okhotoside B_1_ (**8**) at concentrations 1, 2, and 5 μM, (**f**)—okhotoside B_1_ (**8**) at concentrations 0.05, 0.1, and 0.5 μM. ImageJ 1.52 software was used to count the cell colonies. Data are presented as means ± SEM. * *p* value ≤ 0.05 considered significant.

**Figure 3 marinedrugs-22-00560-f003:**
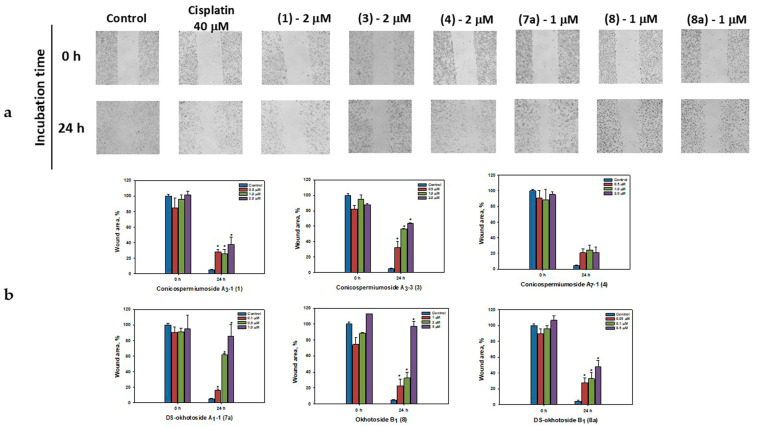
Migration of MDA-MB-231 cells into wound areas observed with an MIB-2-FL fluorescence microscope at 10-fold magnification: (**a**) MDA-MB-231 cells 0 and 24 h without treatment and after treatment with 2 μM of conicospermiumosides A_3_-1 (**1**), A_3_-3 (**3**), A_7_-1 (**4**), and with 1 μM of DS-okhotoside A_1_-1 (**7a**), okhotoside B_1_ (**8**), and DS-okhotoside B_1_ (**8a**); (**b**) MDA-MB-231 cells 0 and 24 h after treatment with cisplatin and 0.5–2 μM of the compounds **1**, **3**, **4**, 0.1–1 μM of the glycoside **7a**, 0.05–0.5 μM of the compound **8a**, and 1–5 μM of the glycoside **8.** Cell migration into wound areas processed by ImageJ 1.52 software. Data are presented as means ± SEM. * *p* value ≤ 0.05 considered significant.

**Figure 4 marinedrugs-22-00560-f004:**
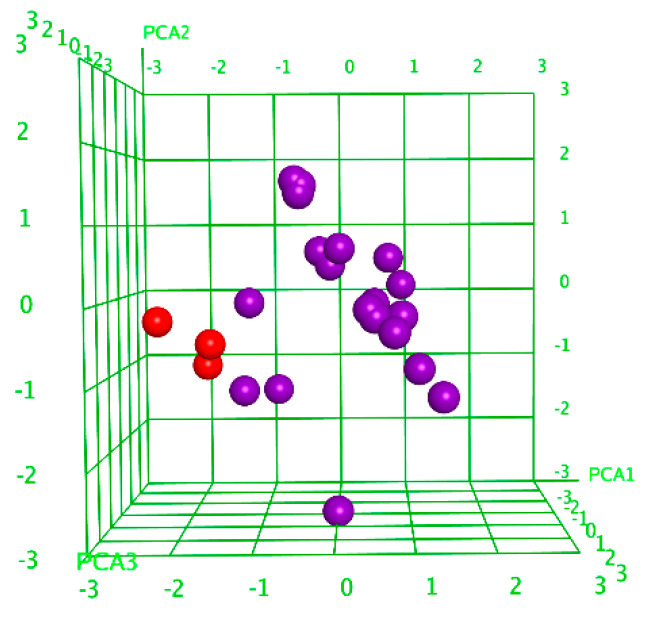
Three-dimensional plot of cytotoxic activity (pIC_50_) depending on the principal components’ values (PCA1–PCA3) calculated for 25 glycosides. The glycosides that demonstrated cytotoxic activity against non-tumor MCF-10A cell line with ID_50_ ≤ 10 µM were outlined as active and are marked in red, while the rest are marked in violet.

**Figure 5 marinedrugs-22-00560-f005:**
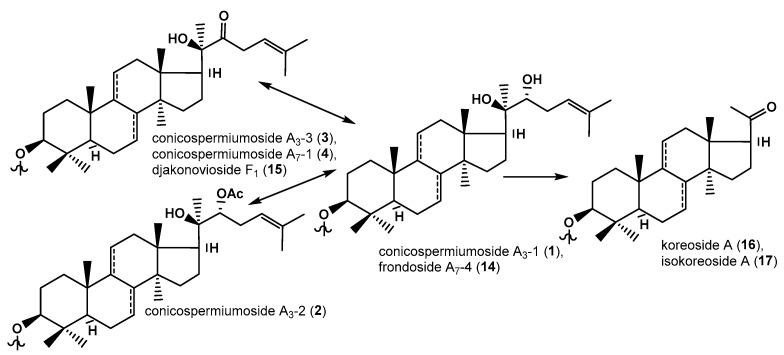
The scheme of biosynthetic transformations of aglycone side chains of the glycosides of *C. conicospermium*.

**Figure 6 marinedrugs-22-00560-f006:**
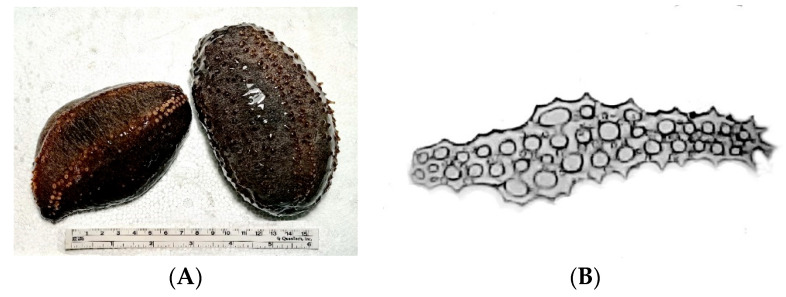
Dorsal and lateral view (**A**) and typical ossicle form of *C. conicospermium* (**B**). Photos by Volod’ko A.V. and Dautov S.Sh.

**Table 1 marinedrugs-22-00560-t001:** ^13^C and ^1^H NMR chemical shifts and the HMBC and ROESY correlations of the carbohydrate moiety of conicospermiumoside A_3_-1 (**1**).

Atom	δ_C_ Mult. ^a,b,c^	δ_H_ Mult. (*J* in Hz) ^d^	HMBC	ROESY
Xyl1 (1→C-3)				
1	104.6 CH	4.72 d (6.9)	C: 3; C: 2 Xyl1	H-3; H-3, 5 Xyl1
2	**81.7** CH	3.95 t (8.8)	C: 1 Qui2; C: 1, 3 Xyl1	H-1 Qui2; H-4 Xyl1
3	75.0 CH	4.28 t (8.8)	C: 2, 4 Xyl1	H-1, 5 Xyl1
4	*76.1* CH	4.97 dd (8.8; 13.9)	C: 3 Xyl1	H-2 Xyl1
5	64.0 CH_2_	4.77 dd (5.7; 12.0)	C: 1, 3 Xyl1	
		3.86 brdd (8.8, 11.8)		H-1, 3 Xyl1
Qui2 (1→2Xyl1)				
1	102.2 CH	5.16 d (7.6)	C: 2 Xyl1	H-2 Xyl1; H-3, 5 Qui2
2	**82.5** CH	3.91 t (9.3)	C: 1, 3 Qui2; C: 1 Xyl5	H-4 Qui2; H-1 Xyl5
3	75.2 CH	3.97 t (9.3)	C: 2, 4 Qui2	
4	**86.4** CH	3.44 t (9.3)	C: 3 Qui2; C: 1 Glc3	H-1 Glc3; H-2 Qui2
5	70.9 CH	3.56 m		H-1, 3 Qui2
6	17.8 CH_3_	1.57 d (6.0)	C: 4, 5 Qui2	
Glc3 (1→4Qui2)				
1	104.0 CH	4.78 d (7.8)	C: 4 Qui2	H-4 Qui2; H-3, 5 Glc3
2	73.6 CH	3.85 t (9.6)	C: 1, 3 Glc3	
3	**86.0** CH	4.18 t (9.6)	C: 2, 4 Glc3, C: 1 MeGlc4	H-1 MeGlc4; H-1, 5 Glc3
4	69.0 CH	3.83 t (9.6)	C: 3, 5, 6 Glc3	H-6 Glc3
5	74.9 CH	4.06 m	C: 1, 3 Glc3	H-1, 3 Glc3
6	67.2 CH_2_	4.92 d (9.6)		
		4.60 dd (6.6; 11.4)	C: 5 Glc3	H-4 Glc3
MeGlc4 (1→3Glc3)				
1	104.5 CH	5.21 d (8.3)	C: 3 Glc3	H-3 Glc3; H-3, 5 MeGlc4
2	74.5 CH	3.85 t (8.3)	C: 1, 3 MeGlc4	
3	86.9 CH	3.67 m	C: 2, 4 MeGlc4; OMe	H-1, 5 MeGlc4
4	70.3 CH	3.89 m	C: 3, 5 MeGlc4	
5	77.5 CH	3.89 m		H-1 MeGlc4
6	61.7 CH_2_	4.34 d (10.7)		
		4.06 brd (2.5; 11.6)	C: 5 MeGlc4	
OMe	60.7 CH_3_	3.80 s	C: 3 MeGlc4	
Xyl5 (1→2Qui2)				
1	105.2 CH	5.18 d (7.7)	C: 2 Qui2	H-2 Qui2; H-3, 5 Xyl5
2	74.9 CH	3.89 t (8.4)	C: 1, 3 Xyl5	
3	76.4 CH	4.05 t (8.4)	C: 2, 4 Xyl5	
4	70.1 CH	4.03 m	C: 3 Xyl5	
5	66.4 CH_2_	4.24 dd (5.4; 11.5)	C: 1, 3, 4 Xyl5	
		3.62 brdd (10.0; 11.5)	C: 1, 3, 4 Xyl5	H-1 Xyl5

^a^ Recorded at 176.04 MHz in C_5_D_5_N/D_2_O. ^b^ Bold—interglycosidic positions. ^c^ Italics—sulfate position. ^d^ Recorded at 700.13 MHz in C_5_D_5_N/D_2_O. Multiplicity by 1D TOCSY. The original spectra of **1** are provided in Appendix A.

**Table 2 marinedrugs-22-00560-t002:** ^13^C and ^1^H NMR chemical shifts and the HMBC and ROESY correlations of the aglycone moiety of conicospermiumoside A_3_-1 (**1**).

Position	δ_C_ Mult. ^a^	δ_H_ Mult. (*J* in Hz) ^b^	HMBC	ROESY
1	35.6 CH_2_	1.33 m		H-3, H-5, H-11
2	27.0 CH_2_	1.95 m		
		1.78 m		H-19, H-30
3	89.0 CH	3.17 dd (4.1; 11.2)	C: 30, 31, C: 1 Xyl1	H-1, H-5, H-31, H1-Xyl1
4	39.5 C			
5	49.5 CH	0.89 d (2.9; 11.7)		H-1, H-3, H-31
6	23.2 CH_2_	1.94 m		
		1.86 m		H-19, H-30
7	122.0 CH	5.65 m	C: 9, 14	H-15
8	148.9 C			
9	48.3 CH	2.30 m		H-18, H-19
10	35.3 C			
11	23.0 CH_2_	1.66 m		H-18
		1.41 m		H-32
12	35.3 CH_2_	1.84 m		H-17
		1.73 m		H-18
13	44.9 C			
14	53.0 C			
15	33.5 CH_2_	1.66 m		
		1.58 m		H-7, H-32
16	22.4 CH_2_	2.22 m		H-22
		1.72 m		H-22
17	51.9 CH	2.05 brt (8.3)	C: 13, 18, 20	H-12, H-21, H-22, H-23, H-32
18	25.0 CH_3_	1.34 s	C: 12, 13, 14, 17	H-9, H-11, H-12, H-19, H-21
19	24.5 CH_3_	0.95 s	C: 1, 5, 9, 10	H-2, H-6, H-9, H-18
20	76.9 C			
21	20.9 CH_3_	1.47 s	C: 17, 20, 22	H-12, H-17, H-18, H-22, H-23
22	77.0 CH	3.73 dd (3.5; 8.6)	C: 20, 21, 24	H-16, H-17, H-21
23	31.6 CH_2_	2.42 m		H-17, H-21
		2.27 m	C: 22, 24, 25	H-21
24	123.1 CH	5.49 brt (7.3)	C: 26, 27	H-26
25	132.2 C			
26	25.8 CH_3_	1.64 s	C: 24, 25, 27	H-24
27	17.8 CH_3_	1.61 s	C: 24, 25, 26	
30	17.4 CH_3_	1.04 s	C: 3, 4, 5, 31	H-2, H-6
31	28.8 CH_3_	1.19 s	C: 3, 4, 5, 30	H-3, H-5, H-6, H-30, H-1 Xyl1
32	30.7 CH_3_	1.07 s	C: 8, 13, 14, 15	H-11, H-12, H-15, H-16, H-17

^a^ Recorded at 176.04 MHz in C_5_D_5_N/D_2_O (4/1). ^b^ Recorded at 700.13 MHz in C_5_D_5_N/D_2_O (4/1). The original spectra of **1** are provided in Appendix A.

**Table 3 marinedrugs-22-00560-t003:** ^13^C and ^1^H NMR chemical shifts and the HMBC and ROESY correlations of the aglycone moiety of conicospermiumoside A_3_-2 (**2**).

Position	δ_C_ Mult. ^a^	δ_H_ Mult. (*J* in Hz) ^b^	HMBC	ROESY
1	35.6 CH_2_	1.32 m		H-3, H-5, H-11
2	26.9 CH_2_	1.96 m		
		1.78 m		H-19
3	88.9 CH	3.17 dd (4.3; 11.6)	C: 1 Xyl1	H-1, H-5, H-31, H1-Xyl1
4	39.4 C			
5	49.4 CH	0.88 dd (3.0; 11.6)		H-1, H-3, H-31
6	23.1 CH_2_	1.94 m		H-31
		1.85 m		H-19
7	120.1 CH	5.64 m	C: 9	H-15, H-32
8	148.9 C			
9	48.2 CH	2.28 brd (14.9)		H-18, H-19
10	35.4 C			
11	23.0 CH_2_	1.67 m		H-1
		1.43 m		
12	35.4 CH_2_	1.84 m		H-17, H-32
		1.73 m		H-18, H-21
13	52.9 C			
14	45.1 C			
15	33.4 CH_2_	1.65 m		H-7, H-12
		1.58 m		
16	22.4 CH_2_	2.31 m		H-22
		1.78 m		H-22, H-32
17	52.9 CH	1.97 brt (7.9)	C: 14, 18	H-12, H-18, H-21, H-22, H-23, H-32
18	24.9 CH_3_	1.33 s	C: 12, 13, 14, 17	H-2, H-9, H-11, H-12, H-19, H-21
19	24.5 CH_3_	0.95 s	C: 1, 5, 9, 10	H-2, H-6, H-9, H-18, H-30
20	75.9 C			
21	21.5 CH_3_	1.52 s	C: 17, 20, 22	H-12, H-17, H-18, H-23
22	79.6 CH	5.19 dd (4.1; 9.2)	C: 20, 21, 24, OAc	H-16, H-17
23	30.1 CH_2_	2.39 m		H-17, H-21
		2.27 brdd (6.7; 14.2)	C: 22, 24, 25	H-21, H-27
24	121.0 CH	5.24 brt (6.7)	C: 26, 27	H-26
25	133.5 C			
26	25.6 CH_3_	1.63 s	C: 24, 25, 27	H-24
27	17.8 CH_3_	1.62 s	C: 24, 25, 26	H-23
30	17.4 CH_3_	1.05 s	C: 3, 4, 5, 31	H-2, H-6, H-19, H-31
31	28.7 CH_3_	1.21 s	C: 3, 4, 5, 30	H-3, H-5, H-6, H-30, H-1 Xyl1
32	30.7 CH_3_	1.07 s	C: 8, 13, 14, 15	H-7, H-12, H-15, H-16, H-17
OCOCH_3_	172.3 C			
OCOCH_3_	23.0 CH_3_	2.10 s	OAc	H-21

^a^ Recorded at 176.04 MHz in C_5_D_5_N/D_2_O. ^b^ Recorded at 700.13 MHz in C_5_D_5_N/D_2_O. The original spectra of **2** are provided in Appendix A.

**Table 4 marinedrugs-22-00560-t004:** ^13^C and ^1^H NMR chemical shifts and the HMBC and ROESY correlations of the aglycone moiety of conicospermiumoside A_3_-3 (**3**).

Position	δ_C_ Mult. ^a^	δ_H_ Mult. (*J* in Hz) ^b^	HMBC	ROESY
1	35.6 CH_2_	1.32 m		H-3, H-11
		1.27 m		
2	27.0 CH_2_	1.94 m		
		1.75 m		H-19, H-30
3	88.9 CH	3.14 dd (4.1; 11.8)	C: 4, 30, 31, 1 Xyl1	H-1, H-5, H-31, H1-Xyl1
4	39.4 C			
5	49.6 CH	0.85 dd (2.4; 11.8)	C: 4, 6, 10, 19, 30	H-3, H-31
6	23.1 CH_2_	1.92 m		
		1.83 m		H-19, H-30
7	122.2 CH	5.62 m	C: 9, 13	H-15
8	148.8 C			
9	48.1 CH	2.28 brd (14.2)		H-18, H-19
10	35.5 C			
11	22.8 CH_2_	1.64 m		H-1
		1.40 m		
12	34.9 CH_2_	1.97 m		H-17
		1.74 m		H-18
13	52.9 C			
14	45.2 C			
15	33.4 CH_2_	1.63 m		
		1.55 m	C: 13, 14, 16	H-7, H-32
16	22.3 CH_2_	2.02 m		H-18, H-21
		1.55 m		
17	53.3 CH	2.37 brt (8.9)	C: 14, 16, 18, 20	H-12, H-21, H-32
18	24.6 CH_3_	1.29 s	C: 12, 13, 14, 17	H-9, H-11, H-12
19	24.5 CH_3_	0.93 s	C: 1, 5, 9, 10	H-2, H-6, H-9
20	81.5 C			
21	24.9 CH_3_	1.62 s	C: 17, 20, 22	H-12, H-16, H-17
22	216.3 C			
23	36.9 CH_2_	3.61 m		H-21
24	117.3 CH	5.45 m	C: 26, 27	H-26
25	134.9 C			
26	25.6 CH_3_	1.64 s	C: 24, 25, 27	H-24
27	17.8 CH_3_	1.57 s	C: 24, 25, 26	
30	17.4 CH_3_	1.03 s	C: 3, 4, 5, 31	H-2, H-6, H-19, H-31
31	28.7 CH_3_	1.18 s	C: 3, 4, 5, 30	H-3, H-5, H-6, H-30, H-1 Xyl1
32	30.7 CH_3_	1.05 s	C: 8, 13, 14, 15	H-11, H-16, H-17

^a^ Recorded at 176.04 MHz in C_5_D_5_N/D_2_O. ^b^ Recorded at 700.13 MHz in C_5_D_5_N/D_2_O. The original spectra of **3** are provided in Appendix A.

**Table 5 marinedrugs-22-00560-t005:** ^13^C and ^1^H NMR chemical shifts and the HMBC and ROESY correlations of the carbohydrate moiety of conicospermiumoside A_7_-1 (**4**).

Atom	δ_C_ Mult. ^a,b,c^	δ_H_ Mult. (*J* in Hz) ^d^	HMBC	ROESY
Xyl1 (1→C-3)				
1	104.6 CH	4.73 d (7.6)	C: 3; C: 5 Xyl1	H-3; H-3, 5 Xyl1
2	**81.6** CH	3.97 t (8.9)	C: 1 Qui2; C: 1, 3 Xyl1	H-1 Qui2; H-4 Xyl1
3	75.1 CH	4.30 t (8.9)	C: 2, 4 Xyl1	H-1, 5 Xyl1
4	*76.2* CH	4.98 dd (8.9; 14.9)	C: 3 Xyl1	H-2 Xyl1
5	64.1 CH_2_	4.79 m	C: 1, 4 Xyl1	
		3.87 dd (8.9; 11.4)		H-1, 3 Xyl1
Qui2 (1→2Xyl1)				
1	102.2 CH	5.18 d (7.5)	C: 2 Xyl1	H-2 Xyl1; H-3, 5 Qui2
2	**82.4** CH	3.94 t (9.6)	C: 1, 3 Qui2; C: 1 Xyl5	H-4 Qui2; H-1 Xyl5
3	75.3 CH	3.98 t (9.6)	C: 2, 4 Qui2	H-1, 5 Qui2
4	86.4 CH	3.46 t (9.6)	C: 3 Qui2; C: 1 Glc3	H-1 Glc3; H-2 Qui2
5	70.9 CH	3.58 m		H-1, 3 Qui2
6	17.9 CH_3_	1.58 d (6.0)	C: 4, 5 Qui2	
Glc3 (1→4Qui2)				
1	104.0 CH	4.79 d (7.9)	C: 4 Qui2	H-4 Qui2; H-3, 5 Glc3
2	73.5 CH	3.81 t (8.8)	C: 1, 3 Glc3	
3	**86.5** CH	4.12 t (8.8)	C: 4 Glc3, C: 1 MeGlc4	H-1 MeGlc4; H-1 Glc3
4	69.1 CH	3.81 t (8.8)	C: 5, 6 Glc3	H-6 Glc3
5	74.9 CH	4.07 m		H-1 Glc3
6	*67.3* CH_2_	4.94 brd (7.9)		
		4.60 dd (5.9; 11.8)	C: 5 Glc3	H-4 Glc3
MeGlc4 (1→3Glc3)				
1	104.8 CH	5.15 d (7.4)	C: 3 Glc3	H-3 Glc3; H-3, 5 MeGlc4
2	74.4 CH	3.77 t (8.2)	C: 1 MeGlc4	H-4 MeGlc4
3	86.4 CH	3.64 t (8.2)	C: 2, 4 MeGlc4; OMe	H-1, 5 MeGlc4
4	69.8 CH	4.01 m	C: 5 MeGlc4	H-2, 6 MeGlc4
5	75.5 CH	4.01 m	C: 6 MeGlc4	H-1, 3 MeGlc4
6	*67.0* CH_2_	4.93 d (11.5)	C: 4, 5 MeGlc4	
		4.74 brd (11.5)	C: 5 MeGlc4	
OMe	60.5 CH_3_	3.76 s	C: 3 MeGlc4	
Xyl5 (1→2Qui2)				
1	105.2 CH	5.21 d (7.4)	C: 2 Qui2; C: 5 Xyl5	H-2 Qui2; H-3, 5 Xyl5
2	74.9 CH	3.91 t (9.6)	C: 1, 3 Xyl5	H-4 Xyl5
3	76.4 CH	4.07 t (9.6)	C: 2, 4 Xyl5	H-1, 5 Xyl5
4	70.1 CH	4.05 m	C: 3 Xyl5	
5	66.4 CH_2_	4.27 dd (5.3; 11.7)	C: 1, 3, 4 Xyl5	H-3 Xyl5
		3.65 t (11.7)	C: 3 Xyl5	H-1, 3 Xyl5

^a^ Recorded at 176.04 MHz in C_5_D_5_N/D_2_O. ^b^ Bold—interglycosidic positions. ^c^ Italics—sulfate position. ^d^ Recorded at 700.13 MHz in C_5_D_5_N/D_2_O. Multiplicity by 1D TOCSY. The original spectra of **4** are provided in Appendix A.

**Table 6 marinedrugs-22-00560-t006:** ^13^C and ^1^H NMR chemical shifts and the HMBC and ROESY correlations of the aglycone moiety of conicospermiumoside A_7_-1 (**4**).

Position	δ_C_ Mult. ^a^	δ_H_ Mult. (*J* in Hz) ^b^	HMBC	ROESY
1	36.3 CH_2_	1.65 m		H-11
		1.28 m		
2	26.7 CH_2_	2.00 m		
		1.77 m		
3	88.7 CH	3.10 dd (4.1; 11.9)	C: 4, 30, 31, 1 Xyl1	H-1, H-5, H-31, H1-Xyl1
4	39.7 C			
5	52.9 CH	0.78 m	C: 6, 10, 19, 30, 31	H-3, H-31
6	21.2 CH_2_	1.60 m		
		1.31 m		
7	28.2 CH_2_	1.53 m		
		1.23 m		
8	41.4 CH	2.09 m		H-18, H-19
9	148.2 C			
10	39.2 C			
11	114.8 CH	5.18 m	C: 8, 10, 13	
12	37.6 CH_2_	2.24 btd (17.0)		
		2.02 m	C: 9, 11, 13, 14	
13	45.5 C			
14	47.0 C			
15	33.7 CH_2_	1.45 m	C: 14, 16, 32	H-18
		1.33 m		
16	22.2 CH_2_	2.01 m		H-18
		1.53 m		
17	51.3 CH	2.54 brt (9.5)	C: 12, 13, 16, 18, 20, 22	H-12, H-21, H-32
18	16.4 CH_3_	1.01 s	C: 12, 13, 14, 17	H-8, H-12, H-15, H-21
19	22.2 CH_3_	0.95 s	C: 1, 5, 9, 10	H-6, H-8, H-30
20	81.6 C			
21	24.6 CH_3_	1.61 s	C: 17, 20, 22	H-16, H-17, H-18
22	216.4 C			
23	36.9 CH_2_	3.64 m		H-21
24	117.3 CH	5.44 m	C: 26, 27	
25	135.0 C			
26	25.6 CH_3_	1.63 d (1.4)	C: 24, 25, 27	H-24
27	17.9 CH_3_	1.56 d (1.4)	C: 24, 25, 26	
30	16.7 CH_3_	0.99 s	C: 3, 4, 5, 31	H-2, H-6, H-31
31	28.1 CH_3_	1.17 s	C: 3, 4, 5, 30	H-3, H-5, H-6, H-30
32	18.8 CH_3_	0.77 s	C: 8, 13, 14, 15	H-7, H-15, H-17

^a^ Recorded at 176.04 MHz in C_5_D_5_N/D_2_O. ^b^ Recorded at 700.13 MHz in C_5_D_5_N/D_2_O. The original spectra of **4** are provided in Appendix A.

**Table 7 marinedrugs-22-00560-t007:** ^13^C and ^1^H NMR chemical shifts and the HMBC and ROESY correlations of the aglycone moiety of conicospermiumoside A_7_-2 (**5**).

Position	δ_C_ Mult. ^a^	δ_H_ Mult. (*J* in Hz) ^b^	HMBC	ROESY
1	35.6 CH_2_	1.33 m		
2	26.9 CH_2_	1.95 m		
		1.78 m		H-19
3	88.9 CH	3.17 dd (3,4; 11.8)		H-5, H-31, H1-Xyl1
4	39.4 C			
5	49.4 CH	0.88 brd (11.8)		H-1, H-3, H-31
6	23.1 CH_2_	1.93 m		H-31
		1.86 m		
7	121.9 CH	5.65 m		
8	148.7 C			
9	48.2 CH	2.29 brd (14.6)		H-18, H-19
10	39.9 C			
11	24.5 CH_2_	1.63 m		
		1.31 m		
12	35.2 CH_2_	1.78 m		
		1.67 m		
13	52.9 C			
14	44.7 C			
15	33.4 CH_2_	1.68 m		
		1.62 m		
16	22.1 CH_2_	2.27 m		
		1.82 m		H-32
17	51.9 CH	2.09 brt (9.1)		H-32
18	25.0 CH_3_	1.35 s	C: 12, 13, 14, 17	H-9, H-11, H-12
19	24.5 CH_3_	0.93 s	C: 1, 5, 9, 10	H-2, H-6, H-9, H-30
20	77.1 C			
21	20.8 CH_3_	1.50 s	C: 17, 20, 22	H-12, H-17, H-23
22	78.1 CH	4.29 m	C: 24, 25	H-24
23	126.2 CH	6.13 m	C: 22	H-17, H-21
24	141.8 CH	6.13 m	C: 25	H-24, H-26
25	69.8 C			
26	29.8 CH_3_	1.49 s	C: 24, 25, 27	H-24
27	30.1 CH_3_	1.48 s	C: 24, 25, 26	
30	17.4 CH_3_	1.04 s	C: 3, 4, 5, 31	H-2, H-6, H-31
31	28.7 CH_3_	1.19 s	C: 3, 4, 5, 30	H-3, H-5, H-6, H-30
32	30.6 CH_3_	1.05 s	C: 8, 13, 14, 15	H-11, H-17

^a^ Recorded at 176.04 MHz in C_5_D_5_N/D_2_O. ^b^ Recorded at 700.13 MHz in C_5_D_5_N/D_2_O. The original spectra of **4** are provided in Appendix A.

**Table 8 marinedrugs-22-00560-t008:** The cytotoxic activities of glycosides of *C. conicospermium* against human erythrocytes (okhotoside A_1_-1 (**7**) is positive control) and the MCF-10A, MCF-7, T-47D, and MDA-MB-231 human cell lines (cisplatin is positive control).

Glycosides	ED_50_, µM, Erythrocytes	Cytotoxicity, IC_50_ µM			
		MCF-10A	MCF-7	T-47D	MDA-MB-231
Conicospermiumoside A_3_-1 (**1**)	0.77 ± 0.07	13.56 ± 0.71	41.67 ± 3.08	19.91 ± 1.02	5.93 ± 0.35
Conicospermiumoside A_3_-2 (**2**)	7.56 ± 0.08	36.80 ± 2.48	>50.00	38.94 ± 1.62	20.67 ± 2.28
Conicospermiumoside A_3_-3 (**3**)	0.21 ± 0.02	11.42 ± 0.58	28.49 ± 1.18	10.68 ± 0.94	5.62 ± 0.26
Conicospermiumoside A_7_-1 (**4**)	0.36 ± 0.04	40.25 ± 2.10	>50.00	19.73 ± 1.01	4.78 ± 0.23
Conicospermiumoside A_7_-2 (**5**)	20.42 ± 0.36	>50.00	>50.00	>50.00	49.87 ± 1.45
Okhotoside A_1_-1 (**7**)	0.42 ± 0.02	7.95 ± 0.44	10.92 ± 0.96	6.23 ± 0.68	2.25 ± 0.17
DS-okhotoside A_1_-1 (**7a**)	0.24 ± 0.02	7.81 ± 0.32	18.81 ± 1.02	7.80 ± 0.31	2.83 ± 0.36
Okhotoside B_1_ (**8**)	1.05 ± 0.12	40.95 ± 0.81	42.91 ± 2.00	20.31 ± 0.85	11.35 ± 0.20
DS-okhotoside B_1_ (**8a**)	0.33 ± 0.03	9.87 ± 0.73	18.82 ± 0.71	9.62 ± 0.30	1.09 ± 0.08
Okhotoside B_2_ (**9**)	1.36 ± 0.10	10.69 ± 0.21	39.81 ± 2.96	11.11 ± 1.15	4.25 ± 0.38
Frondoside A (**10**)	1.22 ± 0.06	11.00 ± 1.04	19.45 ± 0.48	9.76 ± 0.53	2.79 ± 0.09
Frondoside A_7_-4 (**14**)	2.16 ± 0.06	26.80 ± 0.94	32.87 ± 2.13	44.51 ± 2.18	8.11 ± 0.74
cisplatin	-	74.30 ± 2.19	112.92 ± 2.45	>160.00	≥160.00

**Table 9 marinedrugs-22-00560-t009:** The anti-proliferative activity of glycosides of *C. conicospermium* against human cell lines: MCF-10A and MDA-MB-231 at 48 h and 72 h.

Glycosides	Cytotoxicity, IC_50_ µM			
	48 h		72 h	
	MCF-10A	MDA-MB-231	MCF-10A	MDA-MB-231
Conicospermiumoside A_3_-1 (**1**)	30.42 ± 1.05	2.54 ± 0.09	38.98 ± 2.15	12.40 ± 0.75
Conicospermiumoside A_3_-3 (**3**)	10.58 ± 0.72	10.08 ± 0.71	5.27 ± 0.36	5.18 ± 0.43
Conicospermiumoside A_7_-1 (**4**)	35.51 ± 1.59	15.29 ± 1.01	11.08 ± 0.57	11.18 ± 0.73
DS-okhotoside A_1_-1 (**7a**)	5.18 ± 0.28	4.32 ± 0.30	3.87 ± 0.24	2.73 ± 0.18
Okhotoside B_1_ (**8**)	15.54 ± 0.81	11.73 ± 1.00	16.49 ± 0.98	2.92 ± 0.07
DS-okhotoside B_1_(**8a**)	5.81 ± 0.46	2.63 ± 0.14	2.77 ± 0.25	0.60 ± 0.03

**Table 10 marinedrugs-22-00560-t010:** Tumor cell selectivity index (SI; a ratio of IC_50_ calculated for healthy (MCF-10A) and cancer cells (MDA-MB-231)) of tested glycosides.

Glycosides	SI		
	24 h	48 h	72 h
Conicospermiumoside A_3_-1 (**1**)	2.29	11.97	3.14
Conicospermiumoside A_3_-3 (**3**)	2.03	1.05	1.02
Conicospermiumoside A_7_-1 (**4**)	8.47	2.32	0.99
DS-okhotoside A_1_-1 (**7a**)	2.76	1.20	1.42
Okhotoside B_1_ (**8**)	3.61	1.32	5.65
DS-okhotoside B_1_(**8a**)	9.06	2.21	4.62

## Data Availability

The raw data supporting the conclusions of this article will be made available by the authors on request.

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
