# Peer review of "Composition of Triterpene Glycosides of the Far Eastern Sea Cucumber Cucumaria conicospermium Levin et Stepanov; Structure Elucidation of Five Minor Conicospermiumosides A3-1, A3-2, A3-3, A7-1, and A7-2; Cytotoxicity of the Glycosides Against Human Breast Cancer Cell Lines; Structure–Activity Relationships"

_marinedrugs, 2024, doi:10.3390/md22120560_

Round 1

Reviewer 1 Report

Comments and Suggestions for Authors

The manuscript reported the isolation, structure elucidation and bioactivities of five new and 12 known conicospermiumosides isolated from the Far Eastern Sea Cucumber Cucumaria conicospermium Levin et Stepanov. Overall, the authors provided valuable information to natural products research field. However, some major revisions should be done to improve the manuscript.

1.     It will be better to include the first paragraph in 2.1. Structure Elucidation of Glycosides to section 4.3. Extraction and Isolation.

2.     The bioassay data showed that the isolated compounds seemed having moderate cytotoxicity at uM level instead of promising bioactivity. Please consider revising the conclusion in the manuscript.

3.     Although the authors claimed that “The sugar absolute configuration for 1 and all other glycosides was assigned as D as in all other sea cucumber along with biogenetic reasons.”, it does not mean all the new compounds possessed the identical absolute configurations. At least, one compounds should be further studied to provide more evidences. Similarly, the absolute configurations of aglycons should be demonstrated, not just based on the biogenetic reasons.

4.     Section 4.3.1-5: Please check the mp. It should be range, not just an accurate number. For new compounds, the UV, IR, CD should be included if possible.

5.     Consider only limited number of compounds were studied for their bioactivities, the QSAR conclusion should be prelimary.

6.     Figure 5 could be removed, because the biosynthetic pathway is simple.

7.     Please the format errors in the References section.

Author Response

We are very grateful to the Reviewer for his in-depth analysis of our manuscript. The responses to the comments are expressed below:

  1. It will be better to include the first paragraph in 2.1. Structure Elucidation of Glycosides to section 4.3. Extraction and Isolation.

Answer: Since paragraph 2.1 duplicates the data described in section 4.3, it has been largely shortened.

  1. The bioassay data showed that the isolated compounds seemed having moderate cytotoxicity at uM level instead of promising bioactivity. Please consider revising the conclusion in the manuscript.

Answer: Actually, the tested compounds demonstrated cytotoxic action against most sensitive cell line MDA-MB-231 at rather broad range of concentrations (from 1.09 to 49.87 mM), but six selected glycosides were active at the doses 1.09–11.35 mM. The latter value really only indicates moderate activity. But more importantly that such anticancer effects, as anti-colony forming activity and inhibition of cells’ migration were observed at much lower concentrations: conicospermiumosides A3-1 and A3-3 almost completely suppressed the colony growth at maximal concentrations of 5 μM. The lower concentration of 2 μM of conicospermiumoside A3-1 caused inhibition of colony formation and growth by ~40% while the same concentration of conicospermiumoside A3-3 showed the ~97% effect. Moreover, okhotoside A1-1 and its desulfated derivative inhibited this property of cancer cells by ~97% even at concentration of 1 μM.

As regards the suppression of cell motility, some of selected glycosides (DS-okhotoside A1-1 and conicospermiumoside A3-3) demonstrated significant effects (by 61.2% and 63.7% as compared with control) at the doses of 0.5 μM and 2 μM, respectively. The activity became stronger with the concentration increasing. So, the glycosides caused significant suppression of cancer cells in rather low concentrations. That's exactly what the conclusions say.

  1. Although the authors claimed that “The sugar absolute configuration for 1 and all other glycosides was assigned as D as in all other sea cucumber along with biogenetic reasons.”, it does not mean all the new compounds possessed the identical absolute configurations. At least, one compounds should be further studied to provide more evidences. Similarly, the absolute configurations of aglycons should be demonstrated, not just based on the biogenetic reasons.

Answer: Regarding the sugar absolute configurations: the assigning of configurations of monosaccharides of the glycosides of Cucumaria japonica (closely related species to C. conicospermium) was performed earlier on the base of specific rotation value ([α]D20) of the monosaccharides obtained after the acid hydrolysis of the glycosides. The sugars have D-configuration [Avilov S.A.; Stonik V.A.; Kalinovsky A.I. Structure of four new triterpene glycosides from the holothurian Cucumaria japonica. Chem. Nat. Compd. 1990, 26, 6, 670-675. DOI: 10.1007/BF00630079]. Later, in 2015 absolute configuration of monosaccharides was established as D for new glycoside isolated from C. japonica after obtaining of octyl-derivatives of sugars followed by their acetylation and GLC analysis together with authentic samples of corresponding monosaccharides (L and D forms) treated using the same methodology [Silchenko A.S., Kalinovsky A.I., Dmitrenok P.S., Kalinin V.I., Mazeika A.N., Vorobieva N.S., Sanina N.M., Kostetsky E.Y. Cucumarioside E from the Far Eastern sea cucumber Cucumaria japonica (Cucumariidae, Dendrochirotida), new minor monosulfated holostane triterpene pentaoside with glucose as the second monosaccharide residue. Nat. Prod. Commun. 2015, 10, 6, 877–880.]. Moreover, D-configuration of all the monosaccharide residues in the carbohydrate chain of okhotoside A1-1 isolated first from Cucumaria okhotensis and also found by us in C. conicospermium, was established after acidic hydrolysis of native glycoside, the subsequent alcoholysis of the monosaccharide mixture by the action of (R)-(–)-2-octanol, followed by acetylation, and GLC analysis in the presence of octyl-derivatives of corresponding standards [Silchenko, A.S.; Avilov, S.A.; Kalinin, V.I.; Stonik, V.A.; Kalinovsky, A.I.; Dmitrenok, P.S.; Stepanov, V.G. Monosulfated Triterpene Glycosides from Cucumaria okhotensis Levin Et Stepanov, a New Species of Sea Cucumbers from Sea of Okhotsk. Rus. J Bioorg. Chem., 2007, 33, 1, 73–82.].

So, all other sea cucumber species belonging to the orders Dendrochirotida and Apodida for which absolute configurations of monosaccharides comprising triterpene glycosides were determined contained only D-sugars, including the sea cucumbers belonging to the genus Cucumaria. Thus, we suppose the configuration of monosaccharides in the glycosides of C. conicospermium is evolutionarily fixed as D. Given these data we have allowed ourselves to rely on biogenetic prerequisites. Data on the monosaccharides D-configuration of okhotoside A1-1 isolated from C. conicospermium are added to the text.

Regarding the absolute configurations of aglycones: it only made sense to determine the configuration of C-22 by Mosher’s method in new compounds, conicospermiumosides A3-1 and A7-2, since the configurations of all other chiral centers in the polycyclic systems of the aglycones were assigned and elucidated earlier. However, these glycosides were isolated in rather small quantities (4.1 and 1.6 mg) and were necessary for the biological activity testing. Given the fact that the attempt to assign a C-22 configuration using the modified Mosher’s method can be failed as it had happened with djakonoviosides A1 and B1 [Silchenko, A.S.; Kalinovsky, A.I.; Avilov, S.A.; Popov, R.S.; Dmitrenok, P.S.; Chingizova, E.A.; Menchinskaya, E.S.; Panina, E.G.; Stepanov, V.G.; Kalinin, V.I.; et al. Djakonoviosides A, A1, A2, B1–B4—triterpene monosulfated tetra- and pentaosides from the sea cucumber Cucumaria djakonovi: the first finding of a hemiketal fragment in the aglycones; activity against human breast cancer cell lines. Int. J. Mol. Sci. 2023, 24, 11128. https://doi.org/10.3390/ijms241311128.] because the 22-O-MTPA esters were not formed, we have decided to rely on biogenetic background and cited evidences of saving the same orientation of functional groups at C-22 in different aglycones. Thus, configuration of C-22 was assigned as R in cladoloside C (having holostane aglycone with 22-OH group and 25(26)-double bond) by Mosher’s method [Kalinovsky A.I., Silchenko A.S., Avilov S.A., Kalinin V.I. The assignment of the absolute configuration of C-22 chiral center in the aglycones of triterpene glycosides from the sea cucumber Cladolabes schmeltzii and chemical transformations of cladoloside C. Nat. Prod. Commun. 2015, 10, P. 1167–1170.], in frondoside C having lanostane aglycone with 22(R)-O-acetic group [Avilov S.A., Drozdova O.A., Kalinin V.I., Kalinovsky A.I., Stonik V.A., Gudimova E.N., Riguero R., Jimenes C. Frondoside C, a new nonholostane triterpene glycoside from the sea cucumber Cucumaria frondosa: structure and cytotoxicity of its desulfated derivative. Can. J. Chem. 1998, 76, 137–141.] on the base of coincidence of the corresponding chemical shifts of the carbons of this glycoside and the known crustecdysone derivative - 2b,3b,(22R)-triacetoxy-14a,(20R),25-trihydroxy-5b-cholest-7(8)-en-6-one, as well as comparison of NMR spectra of frondoside C with the spectra of four synthetic stereoisomers of 5a-cholestan-3b,20,22-triol by C-20 and C-22. For holostane aglycone of cucumarioside Н8 from E. fraudatrix having (16S,22R)-epoxy-fragment (22R)-configuration was deduced from NOE-correlations [Silchenko A.S., Kalinovsky A.I., Avilov S.A., Andryjaschenko P.V., Dmitrenok P.S., Yurchenko E.A., Kalinin V.I. Structure of cucumariosides H5, H6, H7 and H8. Glycosides from the sea cucumber Eupentacta fraudatrix and unprecedented aglycone with 16,22-epoxy-group // Nat. Prod. Commun. 2011. V. 6. P. 1075–1082.]. The coincidence of configuration of C-22 in all holostane aglycones and non-holostane ones appeared to indicate that they are biosynthesized by the same ways and allowed us to suppose the same configuration of this asymmetric center in the glycosides of C. conicospermium.

  1. Section 4.3.1-5: Please check the mp. It should be range, not just an accurate number. For new compounds, the UV, IR, CD should be included if possible.

Answer: The values of melting points were corrected in line with the Reviewer comment. As we apply modern and precise methods (1D and 2D NMR spectroscopy and HR ESI mass spectrometry) for structure elucidation of the glycosides, the using of IR and UV is uninformative, because IR can detect only the presence of γ-lactone, keto-, acetic- and hydroxy-groups. UV spectroscopy is used rarely for confirmation of some specific chemical features, such as diketone fragment conjugated with double bond or α,β-unsaturated ketone fragment like in two glycosides isolated from Cucumaria fallax [Silchenko A.S., Kalinovsky A.I., Avilov S.A., Adnryjaschenko P.V., Dmitrenok P.S., Kalinin V.I., Martyyas E.A., Minin K.V. Fallaxosides C1, C2, D1 and D2, unusual oligosulfated triterpene glycosides from the sea cucumber Cucumaria fallax (Cucumariidae, Dendrochirotida, Holothurioidea) and a taxonomic stratus of this animal. Nat. Prod. Commun. 2016, 11, 7, 939–945]. But the glycosides of C. conicospermium did not contain such unusual structural fragments, hence no further evidences were necessary. There were no examples of using of CD for the triterpene glycosides structures establishing due to complexity of their molecules.

  1. Consider only limited number of compounds were studied for their bioactivities, the QSAR conclusion should be prelimary.

Answer: Structure-activity relationships of the sea cucumber glycosides has been studied for quite some time and the main tendencies are deduced. However modern computer-aided calculation techniques like correlational analysis and QuaSAR-Descriptor tool of the MOE 2020.0901 CCG software allowed to broaden and refine our knowledge. Recently performed analysis of SAR and QSAR of the broad series of the glycosides from sea cucumbers confirmed the complicated character of these relationships and determined some structural features significantly influencing the activity: the presence of a developed carbohydrate chain composed of four to six monosaccharide residues or a disaccharide chain with a sulfate group;  the availability of 18(20)- or 18(16)-lactone and a normal (non-shortened) side chain; the presence of 9β-H,7(8)-en fragment, or 9(11)-double bond. It was shown that the impact of sulfate groups to the membranotropic action of the glycosides depends on the architecture of the sugar chain and the positions of sulfate groups. Hydroxyl groups attached to different positions of aglycone side chains extremely decrease the activity [Zelepuga, E.A.; Silchenko, A.S.; Avilov, S.A.; Kalinin, V.I. Structure-activity relationships of holothuroid’s triterpene glycosides and some in silico insights obtained by molecular dynamics study on the mechanisms of their membranolytic action. Mar. Drugs 2021, 19, 604. https://doi.org/10.3390/md19110604]. The QSAR analysis of the glycosides of C. djakonovi showed the consistency with the observed structure–activity relationships and confirmed earlier conclusions. Some new details were disclosed: the negative correlation of the molecular volume and shape with the hemolytic activity, that was confirmed by the observation that tetraosides with linear carbohydrate chains showed stronger effects than the corresponding pentaosides; the presence of a third sulfate group, unlike the second one, is not conducive to the membranotropic properties of the analyzed glycosides [Silchenko, A.S.; Kalinovsky, A.I.; Avilov, S.A.; Popov, R.S.; Chingizova, E.A.; Menchinskaya, E.S.; Zelepuga, E.A.; Panina, E.G.; Stepanov, V.G.; Kalinin, V.I.; et al. Sulfated triterpene glycosides from the Far Eastern sea cucumber Cucumaria djakonovi: djakonoviosides C1, D1, E1, and F1; cytotoxicity against human breast cancer cell lines; quantitative structure–activity relationships. Mar. Drugs 2023, 21, 602. https://doi.org/10.3390/md21120602.]. QSAR calculations of the glycosides of Psolus peronii corroborated earlier deduced patterns and added some data, such as the sulfate group at C-2 of the terminal glucose residue in the upper semi-chain caused the electrostatic repulsion of the sulfate groups attached to the third and fourth residues in the bottom semi-chain, which led to the “expansion” of the molecule, resulting in a decrease in hemolytic activity, highly grouped negatively charged patches on the surfaces of glycosides led to their high activity, whereas locally distributed areas of positive and negative charges along their molecular surface decreased activity [Silchenko, A.S.; Kalinovsky, A.I.; Avilov, S.A.; Popov, R.S.; Chingizova, E.A.; Menchinskaya, E.S.; Zelepuga, E.A.; Tabakmakher, K.M.; Stepanov, V.G.; Kalinin, V.I. The Composition of Triterpene Glycosides in the Sea Cucumber Psolus peronii: Anticancer Activity of the Glycosides against Three Human Breast Cancer Cell Lines and Quantitative Structure–Activity Relationships (QSAR). Mar. Drugs 202422, 292. https://doi.org/10.3390/md22070292.].

The main finding of the current research of structure-activity relationships was different contributions of the certain structural elements (sometimes even opposite) to the activity against diverse cells indicating glycosides target different membrane components. Such variability of impacts was reflected in the selectivity of action of the glycosides of C. conicospermium towards certain type of cells. Generally, the QSAR model of the cytotoxic effects of glycosides and correlational analysis revealed good coincidence with the results previously obtained. So, from this viewpoint the results of QSAR are preliminary because new calculations supplement the known data. Nevertheless, we already have sufficient grounds to say about regularities and established patterns.

  1. Figure 5 could be removed, because the biosynthetic pathway is simple.

Answer: we completely agree with Reviewer that biosynthetic pathway is simple but this figure makes it easier for the reader to understand the information. So, we would like to keep the figure in the text.

  1. Please the format errors in the References section.

Answer: corrections are made

Reviewer 2 Report

Comments and Suggestions for Authors

This work describe the isolation and characterizarion of seventeen triterpene pentaosides from sea cucumber C. conicospermium. Five of these compounds represent new structures differing by the side chains of the aglycone structure, the location of the aglycone double bond and/or the quantity of sulfate groups in the glycan moiety. One of these compounds (5) displays a novel aglycone side chain whereas compound 4 differer in the location of the aglycone ring double bond with respect to all the other isolated compounds. These five compounds, in addition to other five characterized previously, as well as two desulfated derivatives, were assayed for hemolytic activity and citotoxicity against breast cancer cell lines. Ten compounds from the series demostrated to be strongly active; in particular, two of them showed selective citotoxic activity against cancer cells without significant toxicicty against healthy cells. Quantitative structure-activity relationships allowed to identify structural features that influcence the observed biological activities.

The introduction to the topic is clear and the references are adequate. The manuscript is clearly written, and the data is presented in a very well-structured manner. The work is well planned, and the methodology and results are clearly described.  The structural analysis was conducted using appropriated methods considering the structural complexity of the isolated compounds (i.e. NMR spectroscopy and high-resolution mass spectrometry). The analysis was thorough and well-supported by the data. The structural information in figures and text is adequate and well represented. Good-quality spectra and tables are provided to support the main text description, and they are consistent with each other. The NMR and MS spectra provided in the supporting information are of excellent quality, demonstrating the high purity of the isolated materials. Figure legends provide the necessary information. The conclusions are supported by the data. Besides the structural characterization of novel compounds, the products isolated during this work present interesting biological activities.

In my opinion, this is nice work that contains original material, offering new structural insights into sulfated glycosides from marine source organisms and supporting their potential in the research of new anticancer drugs.  I consider this work would be of interest for readers of this journal and I suggest accepting it for publication with just minor revisions (see comments below).

MINOR COMMENTS:

Material and Methods section: Please, include information about the NMR spectrometer probe used for the studies, temperature and reference used for the 1H and 13C chemical shifts.

NMR Tables: please check the reported multiplicities for the 1H resonances. For instance, H2-H3 of xylose residues should be reported as “dd” instead of “t” and two values of J couplings reported (even though they can be similar). Also, H4 and H5 should be reported as “ddd” and the reported coupling constants should be consistent with those reported for the corresponding coupled spins (i.e. check reported H4-H3, H4-H5ax, H4-H5eq, and H5ax-H5eq). Likewise, the H2, H3 and H4 resonances of glucosyl/quinovosyl residues should be reported as “dd” instead of “t”, and H6a & H6b of glucose as “ddd” each.

Table 1: please check possible typo in HMBC correlations:

* from H1 of Xyl1 where it reads “C1 of Xyl1” since this doesn’t represent a proton-carbon long-range correlation expected in this kind of experiment (may it correspond to “C2 of Xyl1”?)

* from H5 of Glc5 where it reads “C:1,3 of Xyl3” it should “C:1,3 of Glc3

Could the authors please provide more insights related to the following statements, considering the conformational flexibility around the C17-C20 and C20-C22 bonds in the respective structures:

* Page 6 where it reads “The NOE-correlation H-22/H-16 (Table 2) confirmed R-configuration of C-22.” 

* Page 7 where it reads “The configuration of C-20, C-22 chiral centers was assigned as R,R … and confirmed by H-22/H-16, H-17 NOE-correlations.”

Author Response

We are appreciated to the Reviewer for evaluation of our research work. There are the answers for the minor comments:

  1. Material and Methods section: Please, include information about the NMR spectrometer probe used for the studies, temperature and reference used for the 1H and 13C chemical shifts.

Answer: The corresponding data are added to the text: “Avance III 700 Bruker FT-NMR spectrometer (Bruker BioSpin GmbH, Rheinstetten, Germany) (700.13/176.04 MHz (1Н/13C, 30°C, dC 148.9 resonance of C5D5N for 13C and dH 7.21 resonance of C5D5N for 1H used as the references, BBO probe)”

  1. NMR Tables: please check the reported multiplicities for the 1H resonances. For instance, H2-H3 of xylose residues should be reported as “dd” instead of “t” and two values of J couplings reported (even though they can be similar). Also, H4 and H5 should be reported as “ddd” and the reported coupling constants should be consistent with those reported for the corresponding coupled spins (i.e. check reported H4-H3, H4-H5ax, H4-H5eq, and H5ax-H5eq). Likewise, the H2, H3 and H4 resonances of glucosyl/quinovosyl residues should be reported as “dd” instead of “t”, and H6a & H6b of glucose as “ddd” each.

Answer: Indeed, the multiplicities of H-2 and H-3 of the xylose residues as well as H-2, H-3, H-4 of the quinovose and glucose residues should be “doublet of doublets” due to two adjacent non-equivalent protons for each. However, these signals are observed exactly as triplets (multiplicities of monosaccharide protons are deduced from 1D TOCSY spectra acquired for each sugar residue) since the coupling constants are almost equal, so the two midlines in the ‘dd’ merge into one broadened line, turning the “dd” into triplets. Thus, we cannot deduce two coupling constants and describe in the text exactly the data that we observe (see figure in attached file). The differences in the observed and theorized multiplicities of H-4 and H-5 of xylose residues, those actually should be “ddd” but observed as “dd” with broadened central peaks of the doublets: dC 4.97 dd (J = 8.8; 13.9 Hz) (H-4 Xyl1 of conicospermiumoside A3-1 (1)) and 4.77 dd (J = 5.7; 12.0 Hz) (H-5 Xyl1 of conicospermiumoside A3-1 (1)) are explained by the same reasons.

  1. Table 1:please check possible typo in HMBC correlations:
    1. from H1 of Xyl1 where it reads “C1 of Xyl1” since this doesn’t represent a proton-carbon long-range correlation expected in this kind of experiment (may it correspond to “C2 of Xyl1”?)

Answer: the error is corrected, of course HMBC correlation is H-1 Xyl1/C: 2 Xyl1

  1. from H5 of Glc5 where it reads “C:1,3 of Xyl3” it should “C:1,3 of Glc3

Answer: the error is corrected, of course HMBC correlation is H-5 Glc3/C: 1, 3 Glc3

  1. Could the authors please provide more insights related to the following statements, considering the conformational flexibility around the C17-C20 and C20-C22 bonds in the respective structures:
    1. Page 6 where it reads “The NOE-correlation H-22/H-16 (Table 2) confirmed R-configuration of C-22.” 
    2. Page 7 where it reads “The configuration of C-20, C-22 chiral centers was assigned as R,R … and confirmed by H-22/H-16, H-17 NOE-correlations.”

Answer: The same configuration of C-20 was determined in all types of the aglycones: non-holostane without lactone by the comparison of NMR chemical shifts of the CH3-21 in the aglycones with those in modelling derivatives (20S- and 20R-hydroxy-cholesterol) [Miyamoto T., Togawa K., Higuchi R., Komori T., Sasaki T. Structures of four new triterpenoid oligoglycosides: Ds-penaustrosides A, B, C and D from the sea cucumber Pentacta australis. J. Nat. Prod. 1992, 55, 7, 940–946.] or with 18(16)-lactone [Kalinin V.I., Kalinovskii A.I., Stonik, V.A. Onekotanogenin — A new triterpene genin from the holothurianPsolus fabricii. Chem Nat Compd., 1987, 23, 560–563. https://doi.org/10.1007/BF00598672)] and in holostane type aglycones by the X-ray analysis [Il'in, S.G., Reshetnyak, M.V., Oleinikova, G.K. et al. Crystal and molecular structure of 3β-acetoxy-22S,25-epoxyholosta-7,9(11)-dien-17α-ol. Chem Nat Compd 28, 298–301 (1992). https://doi.org/10.1007/BF00630246]. H-17α-orientation was elucidated in all aglycones by typical NOE-correlations H-17/H-32, as well as by the absence of coupling constant J16/17 in the aglycones with 18(16)-lactone.

22R-configuration was determined in lanostane aglycone of frondoside C from C. frondosa [Avilov S.A., Drozdova O.A., Kalinin V.I., Kalinovsky A.I., Stonik V.A., Gudimova E.N., Riguero R., Jimenes C. Frondoside C, a new nonholostane triterpene glycoside from the sea cucumber Cucumaria frondosa: structure and cytotoxicity of its desulfated derivative. Can. J. Chem. 1998, 76, 137–141.] similar to that of new glycosides from C. conicospermium on the base of coincidence of the corresponding chemical shifts of the carbons of this glycoside and the known crustecdysone derivative - 2b,3b,(22R)-triacetoxy-14a,(20R),25-trihydroxy-5b-cholest-7(8)-en-6-one. The comparison of NMR spectra of conicospermiumoside A3-1 (1) with the spectra of four synthetic stereoisomers of 5a-cholestan-3b,20,22-triol by C-20 and C-22 showed the correspondence of chemical shifts in 1 to those in 5a-cholestan-3b,20R,22R-triol stereoisomer: coincidence of signals of C-20 and C-22 (dC 76.9 and 77.0) and the chemical shift of H-21 at dH 1.47. Then, in holostane aglycone of cucumarioside Н8 from E. fraudatrix (16S,22R)-configuration was deduced from J16/17 value 7.7 Hz and NOE correlations Н-16/Н-32 and H-16/H-22 [Silchenko A.S., Kalinovsky A.I., Avilov S.A., Andryjaschenko P.V., Dmitrenok P.S., Yurchenko E.A., Kalinin V.I. Structure of cucumariosides H5, H6, H7 and H8. Glycosides from the sea cucumber Eupentacta fraudatrix and unprecedented aglycone with 16,22-epoxy-group // Nat. Prod. Commun. 2011. V. 6. P. 1075–1082.]. In the ROESY spectra of conicospermiumosides A3-1 (1) and A3-2 (2) the same H-16/H-22 correlation was observed being an additional confirmation of 22R-configuration.

The corresponding clarifications are added to the text:

Page 6: “The configuration of C-22 asymmetric center was deduced as R based on the coincidence of the values of chemical shifts of C-20 and C-22 in 1 that is characteristic for 5a-cholestan-3b,20R,22R-triol stereoisomer [32]. The same configuration of C-22 was deduced earlier for frondoside C having lanostane aglycone without lactone and 22-O-acetoxy-group [13].”

Page 7: “The configuration of C-20, C-22 chiral centers in 2 was assigned as R,R based on biosynthetic background since the aglycone of conicospermiumoside A3-2 (2) turned out to be identical to that of isofrondoside C found first in C. frondosa [16] and confirmed by H-22/H-16, H-17 NOE-correlations.”

Reviewer 3 Report

Comments and Suggestions for Authors

This review concerns the article type manuscript entitled “Composition of Triterpene Glycosides of the Far Eastern Sea Cucumber Cucumaria conicospermium Levin et Stepanov; structure elucidation of five minor conicospermiumosides A3-1, A3-2, A3-3, A7-1, and A7-2; Cytotoxicity of the glycosides against Human Breast” and submitted to Marine Drugs journal (Manuscript ID: marinedrugs-3342379).

The title of the manuscript is consistent with the main text.

The work is focused on isolation and structure determination of five compounds Conicospermiumosides A3-1 (1), A3-2 (2), A3-3 (3), A7-1 (4) and A7-2 (5).

The compounds contain sterane (lanosterane) skeleton substituted at the position 3 by the pentasaccharide chain (the type of saccharide and junction is explained in Table 1 in the text). The structural difference is the side chain carbon skeleton at the position 17 of sterane. In the position 22 is hydroxy A3-1 (1), acetyloxy A3-2 (2), or oxo group A3-3 (3) and unsaturation at the position 24(25). The compound A7-1 (4) is analogue of A3-3 (3) but with 6-sulfate sodium salt at terminal glucopyranose unit. The compound A7-2 (5) has also 6-sulfate sodium salt at terminal glucopyranose unit, but unsaturation at position 23(24) and hydroxy group at the position 25.

The structures of the compounds are determined by the NMR techniques; 1D (1H, 13C, TOCSY) and 2D (COSY, HSQC, ROESY, НМВС). There are also HR-ESI-MS and ESI-MS/MS spectra.

The spectra are given in the supplementary materials. They are well done and of good quality. The data are also given in the main text in tables.

The main part of the text concerns the determination of the structure.

The work is written clearly, although many detailed information makes it difficult to read.

A comparison with analogues described in the literature is also given.It seems that the main idea of ​​the work is to collect data on such compounds. It should be noted, however, that some efforts have been made to establish structure-activity relationships.

In summary, in my opinion, the work can be considered for publication.

Minor remarks: references numeration - lack of 3-6 (probably inside ref.2).

Author Response

We are grateful to the Reviewer for his appreciation of our work.

Minor remarks: references numeration - lack of 3-6 (probably inside ref.2).

Answer: The references were checked: all them are mentioned in the text:

“As the glycosidic composition in the sea cucumbers is usually rather complex, the attempts of obtaining of pure compounds stimulated the modernization of separation approaches of the mixtures of native metabolites [3].

The early studies of different representatives of the genus Cucumaria (family Cucumariidae, order Dendrochirotida) showed their glycosides characterized by species-specific sets of aglycones and genus-specific set of carbohydrate chains including mono-, di- and trisulfated pentaosides, branched by the second monosaccaride residue [4]. Then the set of sugar moieties inherent for Cucumaria’s glycosides was broadened with mono-, di- and trisulfated tetrasaccharide chains found in the glycosides of C. frondosa [5], C. okhotensis [6], C. djakonovi [7, 8].”

Also, the references 3 – 6 are listed in the corresponding section:

References

  1. Kalinin, V.I.; Silchenko, A.S.; Avilov, S.A.; Stonik, V.A. Progress in the studies of triterpene glycosides from sea cucumbers (Holothuroidea, Echinodermata) between 2017 and 2021. Prod. Commun. 2021, 16, 10. https://doi.org/10.1177/1934578X211053934.
  2. Khotimchenko, Pharmacological potential of sea cucumbers. Int. J. Mol. Sci. 2020, 19, 1342. https://doi.org/10.3390/ijms19051342
  3. Silchenko, A.S.; Avilov, S.A.; Kalinin, V.I. Separation procedures for complicated mixtures of sea cucumber triterpene glycosides with isolation of individual glycosides, their comparison with HPLC/MS metabolomic approach, and biosynthetic interpretation of the obtained structural data. In Studies in Natural Products Chemistry; Rahman, A., Ed.; Elsevier Science B.V.: Amsterdam, The Netherlands, 2022; Volume 72, pp. 103–146. https://doi.org/10.1016/B978-0-12-823944-5.00015-6.
  4. Avilov, S.A.; Kalinin, V.I.; Smirnov, A.V. Use of triterpene glycosides for resolving taxonomic problems in the sea cucumber genus Cucumaria (Holothuroidea, Echinodermata). Syst. Ecol. 2004, 32, 715–733. https://doi.org/10.1016/J.BSE.2003.12.008.
  5. Avilov, S.A.; Kalinin, V.I.; Drozdova, O.A.; Kalinovsky, A.I.; Stonik, V.A.; Gudimova, E.N. Triterpene glycosides of the holothurian Cucumaria frondosa. Nat. Compd. 1993, 29, 216–218.
  6. Silchenko, A.S.; Avilov, S.A.; Kalinin, V.I.; Kalinovsky, A.I.; Dmitrenok, P.S.; Fedorov, S.N.; Stepanov, V.G.; Dong, Z.; Stonik, V.A. Constituents of the sea cucumber Cucumaria okhotensis. Structures of okhotosides B1–B3 and cytotoxic activities of some glycosides from this species. Nat. Prod. 2008, 71, 351–356. doi.org/10.1021/np0705413

Round 2

Reviewer 1 Report

Comments and Suggestions for Authors

The authors have revised the manuscript as per the comments. The new version of manuscript is acceptable for publishing in the journal.